# The Promise of RL for Autoregressive Image Editing

**Saba Ahmadi**[1*]   **Rabiul Awal**[1,2*]   **Ankur Sikarwar**[1,2*]   **Amirhossein Kazemnejad**[1*]
**Ge Ya Luo** [1,2]   **Juan A. Rodriguez**[1,4,6]   **Sai Rajeswar**[1,2,6]
**Siva Reddy**[1,3,6,7]   **Christopher Pal** [1,5,6,7]   **Benno Krojer**[1,3]   **Aishwarya Agrawal**[1,2,7]
[1]Mila – Quebec AI Institute    [2]Université de Montréal    [3]McGill University
[4]École de Technologie Supérieure (ETS)    [5]Polytechnique Montréal    [6]ServiceNow
[7]Canada CIFAR AI Chair

## Abstract

While image generation techniques are now capable of producing high-quality images that respect prompts which span multiple sentences, the task of text-guided image editing remains a challenge. Even edit requests that consist of only a few words often fail to be executed correctly. We explore three strategies to enhance performance on a wide range of image editing tasks: supervised fine-tuning (SFT), reinforcement learning (RL), and Chain-of-Thought (CoT) reasoning. In order to study all these components in one consistent framework, we adopt an autoregressive multimodal model that processes textual and visual tokens in a unified manner. We find RL combined with a large multi-modal LLM verifier to be the most effective of these strategies. As a result, we release **EARL**: **E**diting with **A**utoregression and **RL**, a strong RL-based image editing model that performs competitively on a diverse range of edits compared to strong baselines, despite using much less training data. Thus, EARL pushes the frontier of autoregressive multimodal models on image editing. We release our code, training data, and trained models at `https://github.com/mair-lab/EARL`.

## 1  Introduction

With internet-scale image-text pairs [41] and diffusion models [40, 39, 23], we have seen impressive progress on open-ended image generation in recent years. At this point, the latest text-to-image models can often adhere to detailed prompts that span several sentences [5, 13]. However, synthesizing images from a prompt alone is often not sufficient for end-users and broader ML applications. In reality, a person might want to alter highly specific details in a given image instead of creating one from scratch. Beyond direct user applications, e.g., in the domains of robotics and planning, one might want to "imagine into the future" with an image editing model acting as a simulator [59, 26, 7], e.g., "how does this scene look like if the robot pushes the mug?". In both cases, the model must faithfully preserve all details of the original image while modifying the elements intended for editing.

From a capability perspective, most current editing models [61, 8, 56] cover arguably simpler object and attribute edits (*replace, change color, add, ...*), yet only few works [26, 44] tackle more complex edits that require e.g., action understanding or reasoning (spatial, counting, physical dynamics). From a modeling perspective, a standard recipe to improve editing is to apply supervised fine-tuning (SFT) to a diffusion-based image generation model [8, 61, 56], rarely incorporating more recent post-training methods such as reinforcement learning (RL). A parallel line of work introduces additional bounding-box conditioning, either explicitly provided by the user [31] or implicitly predicted [16], leaving these methods far from an end-to-end solution. Hence, in this paper we ask: **What is the**

---

[*]denotes equal contribution

39th Conference on Neural Information Processing Systems (NeurIPS 2025).

**most effective approach to address both simple and complex edits with a unified end-to-end model?** And specifically, what are the key learning paradigms that can move the field forward?

To this end, we conduct a series of experiments with three different learning paradigms (SFT, RL and chain-of-thought reasoning) and mixes of data. However, diffusion-based models would not directly allow a consistent setup where all training approaches could be plugged in out of the box. For this reason we choose Emu3 [50] as our starting point, a fully autoregressive generative model which was pre-trained on captioning and image generation. Since Emu3 is a unified image and language generation model, we can easily use it to study CoT reasoning and online RL methods such as GRPO [42], on top of SFT. We note that RL for visual generation with diffusion or flow-matching is non-trivial [6, 32]. At the end of our exploration, we arrive at a simple recipe and propose **EARL**: **E**diting with **A**utoregression and **RL**, a fully autoregressive generative model that trains on simple and complex editing data during the SFT and RL post-training stages. Specifically, we find the combination of GRPO with a strong MLLM-based verifier to be the most effective. While variants of CLIP-Score are more commonly used verifiers [30, 32], they often require finetuning on preferences and lack fine-grained understanding [60, 1]. Instead, we identify large MLLMs with fine-grained understanding such as Qwen2.5-VL-72B as effective verifiers for a broad range of edits.

We empirically show that EARL performs well across many types of edits by evaluating on 6 diverse benchmarks in both IID and OOD settings. We achieve better results than prior state-of-the-art models on the OmniEdit [52], AURORA [26], and VisMin [2] benchmarks. Moreover, our method outperforms the strongest prior work Omnigen [56], despite using five times less data, and also surpasses baselines that use a comparable amount of data, such as AURORA [26]. See Fig. 1 for samples from EARL before and after applying our RL recipe, showing improved performance even on challenging tasks that require spatial understanding. Notably, modeling textual and visual generation as one autoregressive stream has emerged as a new exciting paradigm [50, 45, 46, 55], and we push the frontier of such models for the *image editing task*, outperforming prior related work EditAR [38]. Finally, we highlight a surprising finding from our in-depth analysis of different training paradigms: teaching the model to explicitly reason about the intermediate steps (chain-of-thought style reasoning) before generating the actual edit does not seem to improve performance, and sometimes it even hurts. Our contributions are as follows:

1. We release **EARL**: A unified end-to-end editing model that performs well on the whole spectrum of edit types.

2. **EARL** outperforms the strongest open-source diffusion baseline (Omnigen [56]), while also outperforming a more comparable fully autoregressive editing model EditAR [38].

3. A simple yet effective online RL pipeline for training an autoregressive editing model.

4. A systematic analysis of how training objectives interplay and at what stages to bring in simple vs. complex edits. The results interestingly reveal that various CoT reasoning settings do not bring any clear improvements. Our findings also show that complex edits are not beneficial during the SFT stage but are effective in the RL post-training stage.

> **EARL Recipe:** Our results show that autoregressive models, while underexplored for image editing, can be highly competitive. When paired with RL, they surpass strong diffusion-based baselines on both simple and complex edits. This demonstrates the power of combining autoregressive generation with RL for controllable and high-fidelity image editing.

## 2 Related Work

**Image Editing Models.** Enabled by the development of diffusion models with open-ended text-conditioning for image generation [40, 39], image editing models can now receive any text prompt in a similar manner and typically depend on these diffusion models [8, 17]. Research [40, 35] has demonstrated that models such as Stable Diffusion [40] can be zero-shot transformed into an editing model by modifying the sampling procedure [35] or attention maps [22]. To achieve better results, components such as additional input U-Net channels are added to pre-trained image generation models followed by fine-tuning on curated editing data [61, 52, 3]. These training datasets consist of triplets of input image, an edit instruction describing the change, and a ground-truth edited image as output. These triplets rarely occur "naturally" (i.e. only in forums [3]) and therefore need

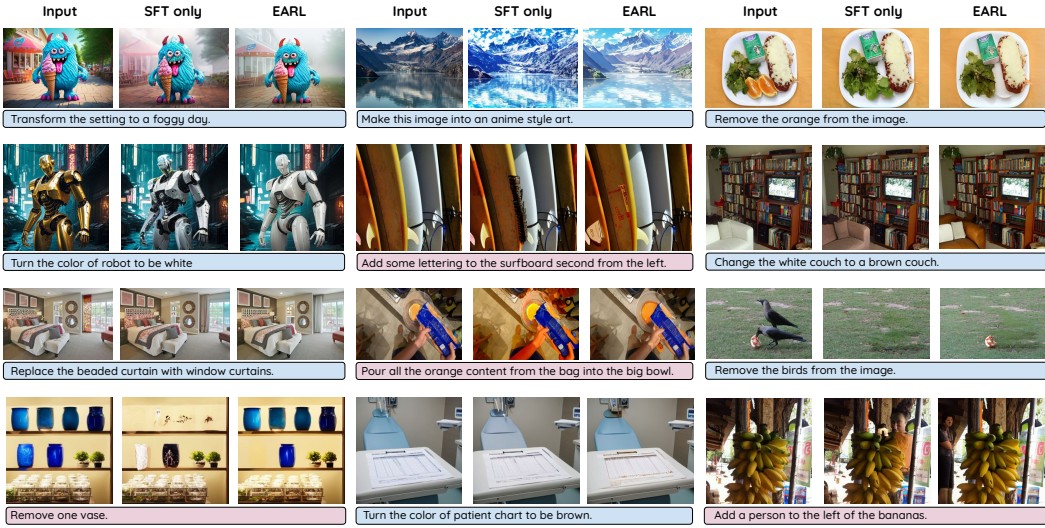

Figure 1: **Qualitative comparison between SFT-only and EARL across diverse editing instructions.** EARL extends the SFT model by leveraging reinforcement learning to better align image edits with natural language prompts. While both models handle simple edits reasonably well, EARL exhibits clear improvements in precise editing on simple as well as complex edit instructions. Simple edit instructions are shown in blue, and complex edit instructions are shown in pink.

to be sourced from synthetic image generation pipelines [22], human-in-the-loop annotation [61], videos [26, 44], or simulation [26, 36]. Various works also adopt more structure into the editing task by restricting the model to edit certain regions of the image [11] or conditioning on bounding boxes or keypoints [37]. In a more recent line of work, image generation is learned jointly with text generation in unified multimodal models that autoregressively predict arbitrary sequences of textual and visual tokens [50, 45]. Concurrent with our work, BAGEL [12] trains a unified transformer model that integrates LLMs and diffusion models, achieving strong editing performance (with and without reasoning) through large-scale pretraining and a powerful base model. However, autoregressive models are less explored for image editing as these models remain less powerful than their diffusion counterparts. Recent work [38] explores using autoregressive models for image editing and achieves competitive results with diffusion baselines. However, they only study the SFT training paradigm, while we study SFT, RL post-training, and CoT Reasoning, cover a wide range of edit types, and outperform their results.

**Reasoning in Image Generation.** LLM reasoning has been adapted to enhance image generation models through additional conditioning or planning [54]. Models like GLIGEN [31] and Layout-GPT [15] use LLMs to predict bounding boxes and scene layouts to direct object placement before generation. In these works, the layout generation step is unimodal, relying solely on LLMs. However, for image editing tasks, incorporating multimodal information from both the original image and the edit instruction is essential to determine an effective layout. GoT (Generation Chain-of-Thought) [14] applies CoT to visual generation and editing tasks. It first generates reasoning in text, analyzing semantic and spatial relationships in the input image, before generating the edited images using a diffusion model. Additionally, PARM++ [21] introduces a reflection mechanism to self-correct generated images, further enhancing the model's reasoning capabilities. Another approach focuses on improving the prompts used for image editing by utilizing a large language model (LLM) or a multimodal LLM (MLLM) [17], which is better equipped for both text and image understanding. While previous work has applied reasoning to image editing tasks, our approach systematically explores and evaluates chain-of-thought (CoT) reasoning in different settings involving both simple and complex edits.

**RL for Image Generation.** Reinforcement learning has emerged as a powerful tool for finetuning image generation models, particularly diffusion models, to better align with human preferences [6].

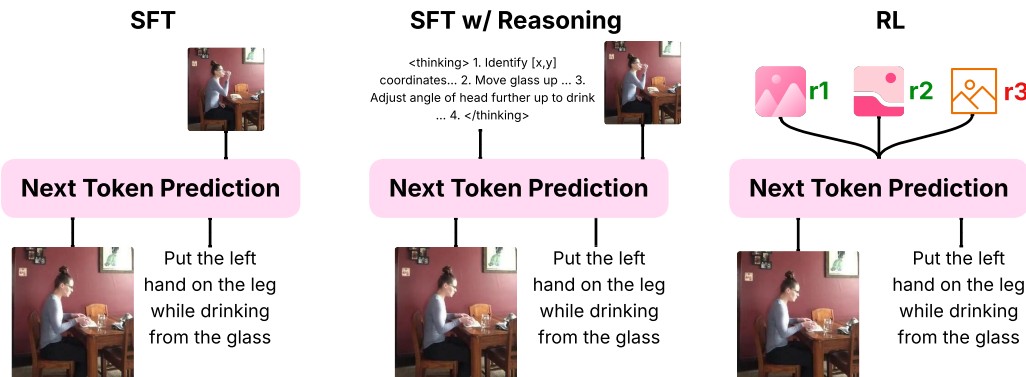

Figure 2: **Autoregressive Image Editing Approaches.** In supervised fine-tuning (SFT), we train an autoregressive model based on the standard image editing setup: triplets of source image, edit instruction, and target image. In SFT with reasoning, the model is supervised to generate chain-of-thought (CoT) reasoning traces before generating the final edited image. Finally, we study reinforcement learning (RL) training of the SFT checkpoint, using edit quality verifiers as reward signals.

Preference-based methods such as Diffusion-DPO [48] and D3PO [57] bypass explicit reward models by directly learning from pairwise human feedback. DDPO [6] further adapts diffusion models to hard-to-specify objectives such as aesthetic quality and compressibility, using rewards based on multimodal models (prompt-image alignment). Although RL for image generation is gaining momentum through human preference and multimodal reward signals, its application to image editing remains underexplored. HIVE [63] collects human feedback on edited images to learn reward functions that capture user preferences, but such datasets are costly and difficult to scale. InstructRL4Pix [30] addresses this challenge by using attention-based reward signals for localized, instruction-driven editing. Meanwhile, GRPO has demonstrated stable and efficient training of autoregressive large language models. Concurrent work such as Flow-GRPO [32] and SimpleAR [49] apply GRPO to flow matching and autoregressive models, respectively. Building on this success, we adopt GRPO for unified image editing with the autoregressive Emu3 model [50]. We leverage a strong multimodal model [58] as a reward function, utilizing its robust image-text alignment and zero-shot prompting to specify and evaluate editing preferences effectively.

## 3   Training Paradigms for Autoregressive Image Editing

To our knowledge, **EARL** is the first to introduce an RL post-training paradigm for *autoregressive* image editing. While RL-based approaches have recently improved diffusion-based editors, their potential in AR models has remained unexplored. Furthermore, we present the first systematic and controlled comparison of three major training strategies: SFT, RL, and CoT reasoning, within a unified AR framework for image editing. In this section, we formally describe the three training paradigms we explore. All our experiments build upon the Emu3 base model [50] (see App. A.1 for more details), a large autoregressive multimodal model that unifies image and text generation in a single autoregressive stream and is trained from scratch.

In the image editing task, the input consists of an original image and a textual edit instruction, and the output is the corresponding edited image. For training Emu3 on the image editing task, both the original and edited images are tokenized into vision tokens using an image tokenizer, while the edit instruction is tokenized into language tokens. These vision and language tokens are then modeled jointly by a single causal transformer (i.e. a LLaMA-style architecture [47]). The task is framed as sequence-to-sequence generation: the model takes as input a token sequence $[x_1, \cdots, x_M]$, formed by concatenating the vision tokens of the input image with the language tokens of the instruction, and generates an output sequence $\hat{y} = [\hat{y}_1, \cdots, \hat{y}_T]$, where $y = [y_1, \cdots, y_T]$ represents the ground truth vision tokens of the edited image which is used for teacher forcing. Below, we detail the learning objective for each of the training paradigms we explore ( Fig. 2):

**Supervised Fine-Tuning (SFT)**   We employ the standard next-token prediction objective to process interleaved image-text sequences. Training minimizes the cross-entropy loss:

$$\mathcal{L}(\theta) = -\mathbb{E}_{(x,y)\sim\mathcal{D}} \left[ \sum_t \log \pi_\theta(y_t \mid y_{<t}, x) \right],$$

where $\pi_\theta$ denotes the model and $\mathcal{D}$ is the labeled training dataset consisting of edit instructions paired with ground-truth edited images.

**Reinforcement Learning (RL) Post-Training**   We use Group Relative Policy Optimization (GRPO) [42] in our RL pipeline to post-train the model following the initial SFT stage. GRPO initializes a trainable policy model $\pi_\theta$ and a frozen reference model from the SFT checkpoint. For a given input prompt $x$, the model generates a group of $G$ responses $y_1, y_2, \ldots, y_G$ based on the current policy model $\pi_{\theta_{\text{old}}}$. The optimization goal is to maximize the following objective:

$$\mathcal{J}(\theta) = \mathbb{E}_{y_1,\ldots,y_G \sim \pi_{\theta_{\text{old}}}} \left[ \frac{1}{G} \sum_{i=1}^{G} \frac{1}{|y_i|} \sum_t \right.$$

$$\left. \min\left( \frac{\pi_\theta(y_{i,t} \mid y_{i,<t})}{\pi_{\theta_{\text{old}}}(y_{i,t} \mid y_{i,<t})} \hat{A}_i, \text{clip}\left( \frac{\pi_\theta(y_{i,t} \mid y_{i,<t})}{\pi_{\theta_{\text{old}}}(y_{i,t} \mid y_{i,<t})}, 1-\epsilon, 1+\epsilon \right) \hat{A}_i \right) - \beta \text{KL}\left[\pi_\theta \| \pi_{\text{ref}}\right] \right].$$

Here, $\hat{A}_i = (r_i - \mu)/\sigma$ denotes the advantage of response $y_i$, where $r_i$ is the reward of the $i$-th response. $\mu$ and $\sigma$ are the mean and standard deviation of rewards across the response group $\{y_i\}$.[2] The hyperparameters $\epsilon$ and $\beta$ control the clipping range and the Kullback–Leibler (KL) penalty, respectively. To compute the reward $r_i$, we first detokenize the vision tokens in $y_i$ back into an image. The reward is then computed using the MLLM verifier, which evaluates the quality of the image editing. This RL objective optimizes the model to generate higher-quality edits while maintaining training stability via the KL divergence term. For a detailed overview of the GRPO method, see the pseudocode in App. A.2.

RL Verifier: Any MLLM with strong image understanding capabilities can be used for verification. In particular, we use Qwen2.5-VL-72B [4] as our verifier to evaluate the generated edits based on the following criteria from VIEScore [27]: (1) *Edit Success* – whether the intended modification was accurately applied; (2) *Overedit* – whether any unintended changes were introduced; (3) *Natural Look* – how well the edit blends with the original image; and (4) *Artifacts* – whether the image contains visual distortions or anomalies. For criteria 1 and 2, the inputs to the model are the edit instruction, input image, and edited image. For criteria 3 and 4, only the edited image is provided. The individual scores are then aggregated into a single reward signal, ranging from 0 to 10.

**Chain-of-Thought (CoT) Reasoning**   Explicitly generating intermediate reasoning steps before the final output is a widely adopted technique that improves performance on complex reasoning tasks [20, 53]. This is known as the Chain-of-Thought (CoT) reasoning. We extend CoT reasoning to the field of image editing. To teach CoT reasoning to Emu3, we finetune Emu3 with CoT supervision by prepending the response $y$ with a tokenized reasoning chain (see Sec. 4.1 for details of synthesizing ground-truth reasoning traces).

## 4   Experimental Setup

### 4.1   Training Details

**Implementation**   Our base model is Emu3-8B [50], a state-of-the-art autoregressive multi-modal model with unified image-text generation capabilities.   For SFT, we initialize from `BAAI/Emu3-Stage1` weights, set a learning rate of $1e-4$, an effective batch size of $128$ with $4$ GPUs, a per-device batch size of $4$, and $8$ gradient accumulation steps. We use validation loss to stop training. For RL post-training, we use a KL divergence coefficient of $3e-4$ and a learning rate of $3e-6$. The RL model is trained with $8$ rollouts per edit instruction and a batch size of $128$ and

---

[2]We omit the dependence on $x$ for brevity.

Table 1: Datasets, covered edit types, and their share of the full training corpus.

| Dataset | Type | Sub-type | Size |
|---|---|---|---|
| **OmniEdit** [52] | S | Object (add, remove, replace), Attribute (modify), Scene, Style | 750K |
| **HumanEdit** [3] | C | Object (add, remove, replace, relation, action counting) | 4.6K |
| **MagicBrush** [61] | C | Object (add, replace, remove), Attribute (modify), OCR (modify), Action (modify) | 8.7K |
| **VisMin** [2] | C | Object (add/remove), Attribute (replace), Count (add/remove), Spatial (swap) | 50K |
| **Aurora-Kubric** [26] | C | Spatial, Counting, Attribute | 50K |
| **Aurora-ActionGenome** [26] | C | Human Pose (action) | 7.8K |
| **SomethingSomething-v2** [19] | C | Action | 50K |

**Editing Data (Input)**

**Input Image** **Edited Image**

**Edit Instruction:** Change the ceramic mug to be painted in a pink color
**Target region:** [[111, 112, 201, 210]]

**Step-by-Step Reasoning (Output)**

**Reasoning:** The source image shows a person holding a painted ceramic mug… The object to be edited is the mug within bounding box [[111, 112, 201, 210]]… Changes include turning the mug color to pink… The edited image will show a person holding a pink painted ceramic mug.

Figure 3: Example of step-by-step reasoning generated by Qwen2.5-VL-72B using standard editing data (input image, edit instruction, target image, and bounding box).

training continues until the reward plateaus. To enhance training stability, we adopt a fully online policy gradient approach, performing a single gradient update at each RL step [25]. All images are resized to $256 \times 256$ by maintaining their original aspect ratio. Further details on the training setup and compute efficiency are provided in App. F.

**Training Datasets** Our dataset is divided into two categories based on the complexity of the edits: Simple Edits (S) and Complex Edits (C). **Simple Edits (S)**: This category includes relatively simple local edits such as single-object and attribute changes, as well as global edits such as style and environment changes. These types of edits are common in large-scale synthetic datasets, such as OmniEdit [52] with $750K$ samples. Existing models generally perform well on these tasks. **Complex Edits (C)**: These edits involve more advanced operations, including counting, spatial, and action modifications, where current models often struggle. Datasets like Aurora-AG [26], Aurora-Kubric [26], VisMin [2], and Something-Something v2 [19] contain such challenging edits. Additionally, we use real-world edit requests curated with human-in-the-loop guidance, such as Human-Edit [3] and MagicBrush [61], which include complex object/attribute changes. Data in the complex edit category is significantly scarcer, e.g., MagicBrush has $8K$ samples.

We provide details for each dataset in Tab. 1. Our dataset S comprises $750K$ OmniEdit samples, while C combines the above mentioned C datasets with $171K$ unique samples. For datasets in C with fewer than $50K$ samples, we upsample them to $50K$, resulting in a dataset of size $300K$ for supervised fine-tuning. For RL post-training, we randomly sample from the respective data pool (S or C) at each iteration, using 16 unique samples per step with 8 rollouts per sample. We first experiment with a smaller setup using a pool of 1,600 samples for various ablations. To improve further, we then train with a total of $32K$ samples over the course of training. This provides more diverse data, allowing the model to benefit from a broader coverage across edit types.

For CoT reasoning supervision, we generate chain-of-thought data using a multimodal large language model (MLLM), Qwen2.5-VL-72B [58], following a prompting strategy similar to [14]. The input to the MLLM consists of standard editing data: an input image, an edited image, a textual edit instruction describing the desired change, and bounding boxes specifying the edit region (if available). For action edits, we also include person keypoints. The generated CoT data follows a step-by-step structure, including a description of the input image, bounding box coordinates of objects to be edited in the input image, bounding boxes of objects to be added in the target image, the edit action, and a description of the final edited image. For complex edit datasets, we apply few-shot prompting to

synthesize CoTs, while for the simple edit dataset (S), we reuse CoTs from [14]. An example is shown in Fig. 3, and our prompting templates and full examples are provided in the App. B.2.

## 4.2 Evaluation Setup

We evaluate on a diverse suite of image editing tasks, ranging from simple object, attribute, and style modifications to complex editing tasks such as counting, actions, and spatial relations.

**Evaluation Metric**   We adopt VIEScore [27] as our metric since it outperforms traditional metrics like LPIPS [62] in terms of human correlation (0.3821 for GPT-4o versus 0.1142 for LPIPS [27]). VIEScore scores edits from 0 to 10 across four criteria (as explained in Sec. 3). For evaluation, we use GPT4o-mini due to its high quality and cost efficiency and confirm that GPT4o-mini variant aligns with human judgment not only on edits from the original human study [27] but also on various complex edits; details are provided in App. C. Note that while our metric and reward are both based on VIEScore, we use separate MLLMs for evaluation (GPT4o-mini) and RL verifier (Qwen2.5-VL-72B), to reduce the risk of metric hacking, i.e. overfitting to MLLM-specific biases.

**Evaluation Benchmarks**   We benchmark on datasets covering both simple (OmniEdit [52], EmuEdit [43]) and complex edits (MagicBrush [61], Aurora [26], and I2EBench [34]). EmuEdit and I2EBench serve as out-of-distribution (OOD) evaluations, with I2EBench including unseen edit types such as lowlight enhancement. We also repurpose VisMin [2], originally an image understanding benchmark, into an editing benchmark by generating edit instructions from captions (App. B.2.2). These datasets span a wide range of edit types and difficulty levels, enabling robust evaluation. To manage VIEScore API costs, we use a 1000-sample subset for I2EBench and EmuEdit.

**Baselines**   We compare our model to diffusion-based baselines, including MagicBrush [61], InstructPix2Pix [8], Aurora [26], and Omnigen, which is the SOTA image editing model [56]. We also compare with EditAR [38], which, to the best of our knowledge, is the only fully autoregressive image editing model.

# 5   Results

## 5.1   Teaching Emu3 Image Editing with Supervised Fine-Tuning

**Simple Editing**   The first row of the Supervised Fine-Tuning section in Tab. 2 presents the results of SFT trained on simple data (SFT (S)). It achieves the highest score on OmniEdit (5.73) and an average score of 3.88, outperforming MagicBrush (3.32) and InstructPix2Pix (3.26), but underperforming Aurora (4.17), EditAR (4.20), and Omnigen (4.70). Notably, Omnigen benefits from a stronger base model with superior image generation performance (GenEval [18] 0.70) and large-scale finetuning ($\sim$ 4M image editing samples). In contrast, our model finetunes from a weaker base model (Emu3, GenEval 0.64) and uses significantly less data ($750K$ image editing samples), resulting in lower overall performance. Also, we see that all models, including Omnigen, perform significantly worse on complex editing benchmarks compared to simple editing benchmarks, **highlighting challenges in spatial edits, changes in object count, and human actions.** Our trained SFT (S) follows this trend. We next explore how to improve SFT performance on both simple and complex edits.

**Complex Editing**   To improve the ability to handle complex edits, we explore two SFT strategies: joint training on the combined simple and complex edit data (SFT (S+C)) and a two-stage curriculum that first finetunes on simple edits, then on complex edits (SFT (S+C) two-stage). As shown in Tab. 2, joint training (SFT (S+C)) reduces average performance compared to simple-only finetuning (SFT (S)) from 3.88 to 3.32 across both simple and complex editing tasks. In particular, the performance drop is significant on simple edit benchmarks, dropping from 5.73 to 4.64 on OmniEdit and from 3.66 to 2.89 on EmuEdit. We hypothesize that this degradation is due to the large distributional shift between simple and complex edits; mixing them early in training may hinder the model's ability to generalize across either. In contrast, the two-stage curriculum partially recovers average performance (3.69) and improves results on some complex edit benchmarks (e.g., VisMin, MB). This suggests that allowing the model to first acquire basic editing capabilities from simple data makes subsequent finetuning on complex tasks more effective, especially given that Emu3 has not been exposed to any

Table 2: SFT and RL model variants for image editing fine-tuning and post-training respectively. S stands for data used in simple editing types, and C stands for data from complex editing types. † and ‡ denote Simple and Complex Edit benchmarks, respectively. $^*$ denotes the best-performing prior state-of-the-art method, a diffusion-based model trained from scratch using approximately ~5x data. Bold numbers indicate the best performances across all methods. Green numbers indicate the performance gain of EARL compared to the SFT (S) baseline.

| Model/Data | Base Model | OmniEdit† | EmuEdit† | AURORA‡ | MB‡ | VisMin‡ | I2EBench‡ | AVG |
|---|---|---|---|---|---|---|---|---|
| Magicbrush | SD v1.5 | 3.43 | 3.28 | 3.01 | 3.64 | 3.48 | 3.06 | 3.32 |
| InstructPix2Pix | SD v1.5 | 3.97 | 3.24 | 3.05 | 3.12 | 2.94 | 3.23 | 3.26 |
| Aurora | SD v1.5 | 4.50 | 4.40 | 4.12 | 4.62 | 3.82 | 3.58 | 4.17 |
| EditAR | LlamaGen | 5.29 | 3.88 | 3.79 | 3.84 | 4.54 | 3.84 | 4.20 |
| Omnigen* | - | 5.68 | **5.00** | 4.10 | **4.68** | 4.09 | **4.68** | 4.70 |
| **Supervised Fine-tuning** | | | | | | | | |
| SFT (S) | Emu3 | 5.73 | 3.66 | 3.58 | 3.19 | 3.57 | 3.59 | 3.88 |
| SFT (S+C) | Emu3 | 4.64 | 2.89 | 2.81 | 2.89 | 3.91 | 2.77 | 3.32 |
| SFT (S+C) two-stage | Emu3 | 4.23 | 3.29 | 3.60 | 3.40 | 4.56 | 3.07 | 3.69 |
| **RL Post-training** | | | | | | | | |
| SFT (S) → RL (S) | Emu3 | 6.07 | 4.13 | 3.47 | 3.53 | 3.34 | 3.80 | 4.06 |
| SFT (S) → RL (C) | Emu3 | 5.94 | 4.12 | 3.84 | 3.92 | 4.09 | 3.90 | 4.30 |
| SFT (S) → RL (S+C) | Emu3 | 6.33 | 4.28 | 3.99 | 4.26 | 4.48 | 4.08 | 4.57 |
| SFT (S+C) → RL (C) | Emu3 | 4.89 | 3.80 | 3.21 | 3.86 | 4.71 | 3.26 | 3.95 |
| SFT (S+C) → RL (S+C) | Emu3 | 5.70 | 4.09 | 3.97 | 4.35 | **4.97** | 3.84 | 4.49 |
| SFT (S+C) two-stage → RL (C) | Emu3 | 4.21 | 3.16 | 3.05 | 3.33 | 4.16 | 2.99 | 3.48 |
| SFT (S+C) two-stage → RL (S+C) | Emu3 | 5.29 | 3.89 | 3.85 | 4.20 | 4.70 | 3.56 | 4.25 |
| **RL Post-training Scaled** | | | | | | | | |
| **EARL SFT (S) → RL (S+C)** | Emu3 | **6.39** | 4.47 | **4.27** | 4.52 | 4.93 | 4.19 | **4.80** |
| Δ **EARL *SFT (S) → RL (S+C) − SFT (S)*** | - | +0.66 | +0.81 | +0.69 | +1.33 | +1.36 | +0.60 | +0.92 |

editing data during pretraining. Overall, the SFT results show that supervised finetuning is insufficient to effectively learn complex editing tasks.

## 5.2   Pushing Image Editing with RL Post-training

In this section, we present results showing that RL post-training substantially improves image editing performance. Starting from the SFT (S) model trained only on simple edits, we apply RL post-training under three settings: RL (S), which uses only simple edit data; RL (C), which uses only complex data; and RL (S+C), which uses both. Tab. 2 shows that **all RL variants outperform the SFT baseline.** RL (C), trained on a disjoint set of complex edits, outperforms RL (S) across all complex (C) benchmarks *without a significant drop on simple (S) benchmarks*. This contrasts with the SFT setting, where adding complex data degraded simple-edit performance. These results indicate that incorporating complex data during RL helps the model learn complex editing while preserving performance on simple edits. The largest gain comes from **balancing simple and complex data, i.e., RL (S+C)**. The best-performing setup, SFT(S) → RL(S+C), improves the average score from $3.88$ to $4.57$, surpassing MagicBrush, InstructPix2Pix, Aurora, and EditAR, and remaining competitive with Omnigen. As shown in Tab. 11, even best-of-5 sampling from the SFT model cannot match RL performance, confirming the gains stem from policy optimization. In addition to accuracy, we analyze inference efficiency in App. F.2: at $256 \times 256$, EARL runs about $4\times$ faster than the SOTA Omnigen with similar quality, while remaining $2\times$ slower than lightweight editors.

Applying RL to models pre-trained on both simple and complex data (SFT (S+C) and two-stage) yields modest gains over SFT, with average scores up to $4.49$, still below RL on the simple-only base (SFT (S) → RL (S+C)) . Fig. 4 shows stable RL training of SFT(S) → RL(S+C) with consistent reward improvements and rising VIEScore on OmniEdit, indicating healthy learning dynamics. The two-stage SFT (S+C) → RL (C) variant reaches only $3.48$, far below SFT (S) → RL (S). Because SFT (S+C) variants underperform relative to SFT (S), their RL counterparts also remain weaker. We hypothesize that supervised finetuning on complex edits may degrade the base model's core capabilities, limiting RL's ability to recover or improve performance. This aligns with findings in LLM research, where base model capability critically influences RL finetuning success [20]. **Thus, while complex data is essential, naively including it during SFT can constrain RL's effectiveness.**

**Scaling RL Training with More Steps and Data**   To further improve performance, we scale the best-performing setup: SFT(S) → RL(S+C), by increasing the duration of RL training to 2000 steps and using a larger data pool of $300K$ (S+C) samples. At each step, 16 unique examples are sampled,

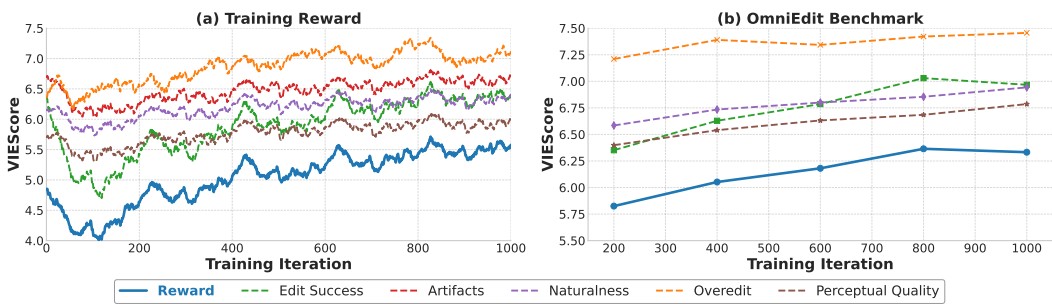

Figure 4: (a) Training curves showing the reward progression, with different aspects of reward. and (b) VIEScore on OmniEdit increases with RL training iterations

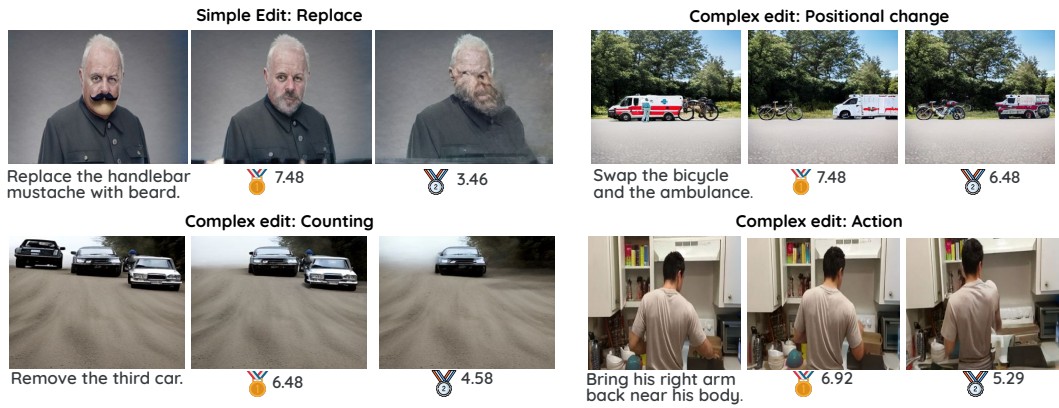

Figure 5: Example reward scores for various types of image edits using Qwen2.5-VL-72B. Higher scores reflect better alignment with the intended edit prompt.

resulting in 32k total samples seen ($20\times$ more than the 1.6k samples from earlier runs). This scaled configuration achieves **the best overall results and is referred to as EARL** in our evaluations. As shown in the last section of Tab. 2, EARL surpasses all baselines (MagicBrush, InstructPix2Pix, Aurora, EditAR, and the SOTA Omnigen), achieving an average score of $4.80$ compared to $4.70$ for Omnigen. In particular, on OmniEdit, AURORA, and VisMin benchmarks, EARL achieves the best results over prior models. We also achieve strong performance on the out-of-distribution benchmarks like I2EBench and EmuEdit.

**Qualitative Analysis of RL Post-training** We observe that the SFT model sometimes produces noisy artifacts or over-edits the surrounding regions. It also fails to achieve successful edits consistently. However, when sampled multiple times, at least one output usually satisfies the correct edit (see examples in Fig. 5). Our verifier captures key aspects of edit quality: edit success, over-editing, presence of artifacts, and naturalness, to guide RL training toward more reliable outputs. Qualitative inspection shows artifacts are nearly eliminated, edits are more precise, and success rates improve after applying RL. Fig. 5 illustrates instances where the verifier successfully handles both simple and complex edit tasks. Nevertheless, the reward model is limited by the multimodal image-text understanding of Qwen2.5-VL-72B. For example, in complex edits, such as changing a higher counts (e.g., six to four), the reward becomes less reliable (see App. D.1 for more details). This also explains why the performance is higher on simple edits than complex edits in EARL. Importantly, a strong reward model is crucial for effective RL. Our 72B verifier demonstrates significantly better alignment with human judgment than the smaller 7B counterpart (see App. D.2 and App. D.3). To better assess EARL's robustness and remaining limitations, we report CLIP similarity and FID metrics in App. E.2, provide fine-grained qualitative analyses in App. E.5, and include a detailed comparison with the only autoregressive baseline EditAR [38] in App. E.3.

## 5.3 Studying Chain-of-thought Reasoning for Editing

Table 3: Performance of EARL variants with chain-of-thought supervision. † and ‡ represents Simple and Complex Edit benchmarks, respectively.

| Model/Data | OmniEdit† | EmuEdit† | AURORA‡ | MB‡ | VisMin‡ | I2EBench‡ | AVG |
|---|---|---|---|---|---|---|---|
| SFT (S) | 5.73 | 3.66 | 3.58 | 3.19 | 3.57 | 3.59 | 3.88 |
| SFT think (S) | 4.34 | 3.76 | 2.88 | 3.36 | 3.46 | 3.21 | 3.50 |
| SFT think (S+C) two-stage | 1.44 | 1.41 | 1.03 | 1.58 | 2.45 | 1.20 | 1.52 |
| SFT think (S) → RL (S) | 4.99 | 3.73 | 3.33 | 3.48 | 3.11 | 3.46 | 3.68 |
| SFT think (S) → RL (C) | 4.36 | 3.67 | 2.94 | 3.59 | 3.08 | 3.16 | 3.47 |
| EARL  SFT think (S) → RL (S+C) | 4.65 | 3.78 | 3.23 | 3.67 | 3.39 | 3.36 | 3.68 |

We evaluate the effect of incorporating chain-of-thought reasoning supervision on image editing performance. Since Chain-of-Thought (CoT) reasoning involves alternating between generating text and images, it requires interleaved image-text generation. However, our base model Emu3 was not pretrained for this type of multimodal generation, which presents additional challenges.

We compare two SFT variants with CoT reasoning supervision: (1) SFT think (S), and (2) SFT think (S+C) using a two-stage approach. Results in Tab. 3 show that, despite the success of chain-of-thought reasoning in large language models [20], SFT think (S) (3.50 avg.) does not improve visual editing performance compared to SFT (S) (3.88 avg.). We also see that **adding complex reasoning data (C) during SFT hurts performance,** as shown by the drop in performance in the SFT think (S+C) two-stage setup compared to SFT think (S). This is consistent with Section 5.1, where including complex reasoning data (SFT (S+C)) and the two-stage (SFT (S+C) two-stage) led to performance degradation compared to SFT (S). Next, we apply RL on top of SFT think (S), using the RL(S), RL (C), and RL (S+C) variants. We leave out applying RL on top of SFT think (S+C) two stage as it is too weak. We perform RL post-training for up to 2,000 steps, or until divergence, using 16 unique samples per step. Applying RL on SFT think (S) yields a slight improvement across two settings: RL (S) and RL (S+C). On average, RL (S) improves performance from 3.50 (SFT (S)) to 3.68, while RL (S+C) also improves from 3.50 (SFT (S)) to 3.68. These observations align with findings in the LLM literature. First, CoT reasoning tends to help only once a model surpasses a capability threshold [10, 53]. Second, RL provides limited benefit when the base model is still sub-optimal [33].

**Quality of Reasoning Chains** The SFT think model generates plausible reasoning: correctly identifies target regions, plans edits, and describes intended outcomes. However, its final outputs often show lower edit accuracy, reduced naturalness, and more artifacts compared to the standard SFT model (see App. E.6 for qualitative examples). These results suggest that although the model learns to generate appropriate reasoning, it does not effectively apply it during generation of the edited image. We hypothesize that this limitation is due to the model's lack of pretraining on interleaved image-text-image data. While the model can generate plausible reasoning chains, it struggles to integrate them effectively for image enhancement, likely because it was not trained to integrate these modalities in a cohesive manner. We leave further exploration of this issue to future work.

## 6 Conclusion

This work delivers the first systematic comparison of supervised finetuning, reinforcement learning, and CoT reasoning for text-guided image editing within a unified autoregressive framework. Directly motivated by this analysis, we introduce a novel autoregressive image editing model, EARL. EARL performs on par with the strongest open-source baselines while using less data, and sets a new bar for multimodal autoregressive models for editing. While SFT alone proves insufficient for handling complex edits, we find that RL significantly improves performance, enhancing the model's overall edit success and ability to handle tasks involving spatial reasoning and dynamic interactions. We also address the question of when to introduce complex edits during the various training stages: we found that bringing in complex edits is not helpful during the SFT stage, but is beneficial during the RL post-training stage. Lastly, the CoT reasoning supervision experiment did not lead to consistent improvements, highlighting the need for further research and stronger autoregressive base models with strong reasoning capabilities. We discuss limitations and broader impact of our work in  App. G.

# 7 Acknowledgments

We acknowledge the valuable feedback provided by Qian Yang, Le Zhang, and Oscar Manas on an early draft of the paper. The technical support extended by the Mila IDT and TamIA teams in managing the computational infrastructure is greatly appreciated. During this project, Aishwarya Agrawal was supported by the Canada CIFAR AI Chair award.

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

# Overview of Appendix

## A  Background

### A.1  Emu3

Emu3 [50] (8B parameters) is an autoregressive multimodal model. It extends the LLaMA-2 [47] architecture by integrating a vision tokenizer (SBER-MoVQGAN) for encoding images into discrete tokens and a text tokenizer (QwenTokenizer) for processing textual inputs. Unlike diffusion-based models [56], Emu3 generates text and visual tokens in a unified manner.

In this work, we use the base version of Emu3, which is pre-trained exclusively on text and image data. During pre-training, Emu3 formats image-text multimodal inputs in a structured format (shown below), incorporating special tokens to distinguish between the different modalities:

"**[BOS]** {caption text} **[SOV]** {resolution info} **[SOT]** {vision tokens} **[EOV] [EOS]**"

where **[BOS]** and **[EOS]** mark the beginning and end of the sequence, **[SOV]** and **[EOV]** define the boundaries of image metadata such as resolution, image tokens, and **[SOT]** indicates the start of vision tokens. Additionally, **[EOL]** and **[EOF]** are included within vision tokens to denote line and frame breaks, respectively. The **resolution info** section contains relevant details to image resolution.

### A.2  GRPO: Group Relative Policy Optimization

---

**Algorithm 1:** Group Relative Policy Optimization

**Input:** initial policy model $\pi_{\theta_{\text{init}}}$; reward models $r_\phi$; task prompts $\mathcal{D}$; hyperparameters $\epsilon$, $\beta$, $\mu$
**Output:** final policy model $\pi_\theta$
**Function** `Group Relative Policy Optimization`:

> $\pi_\theta \leftarrow \pi_{\theta_{\text{init}}}$ ;
> $\pi_{\text{ref}} \leftarrow \pi_{\theta_{\text{init}}}$ ;
> **for** *step = 1 **to** M* **do**
>> Sample a batch $\mathcal{D}_b$ from $\mathcal{D}$ ;
>> Update the old policy model $\pi_{\theta_{\text{old}}} \leftarrow \pi_\theta$ ;
>> Sample $G$ outputs $\{o_i\}_{i=1}^{G} \sim \pi_{\theta_{\text{old}}}(\cdot|q)$ for each question $q \in \mathcal{D}_b$ ;
>> Compute rewards $\{r_i\}_{i=1}^{G}$ for each sampled output $o_i$ by running $r_\phi$ ;
>> Compute $\hat{A}_{i,t}$ for the $t$-th token of $o_i$ through group relative advantage estimation ;
>> **for** *GRPO iteration = 1 **to** $\mu$* **do**
>>> Update the policy model $\pi_\theta$ by maximizing the GRPO objective ;

---

Algorithm 1 presents the GRPO algorithm, where we prepare the advantages using the current policy to estimate the gradients. As noted in Sec. 4, we set the backward batch size equal to the rollout batch size, which effectively reduces the second loop to a single iteration. This means policy gradients are estimated in a fully online manner, with advantages always computed from the current policy. This design choice contributes to improved stability during training.

# B  Training Datasets

## B.1  Dataset Composition

Our goal of building a unified model requires exposure to a wide range of edit types – both simple (e.g., object, attribute, or style changes) and complex (e.g., action changes, counting changes, or spatial relations). No single dataset offers comprehensive coverage of all these edit types, and existing models are typically developed in a fragmented manner, each targeting only a subset of edits.

To address this, we consolidate several existing datasets to form a unified training pool that spans the full spectrum of edit types. *OmniEdit* [52] serves as our largest source of simple editing examples. For complex edits involving actions, relations, counting, and other real-world applications which are significantly less represented than simple edits – we incorporate data from *VisMin* [2], *Aurora* [26], *Human-Edit* [3], and *MagicBrush* [61].

## B.2  Creating Chain-of-Thought from Existing Editing Datasets

This section describes the chain-of-thought datasets for image editing and the associated prompt designs. Image editing datasets typically lack reasoning chains. To address this, we synthesize reasoning chains using a Multimodal Large Language Model (MLLM) through in-context learning (*prompting*), based on existing resources such as the input image, edited image, and edit instruction. We use `Qwen2.5-VL-72B` [58] as the MLLM.

In diffusion models, masking is a common form of conditioning. To mimic this, we collect bounding boxes – either by using those provided in the dataset or by estimating them from available masks and pixel differences. These bounding boxes serve as conditioning inputs in the reasoning chain. The model analyzes the image, edit instruction, and target region (via the bounding box) to generate a step-by-step reasoning process, referred to as the *thinking* field. The system prompt and annotated few-shot examples are provided in App. B.2.1 and App. B.2.3 respectively.

For VisMin, we use image-text pairs from the VisMin [2] dataset, where the two images differ minimally – making them well-suited for editing tasks. Since examples of complex edits, such as those involving counting or spatial relations, are scarce, repurposing VisMin provides useful coverage for these cases. The change can be inferred from the two image captions, which we use to guide `Qwen2.5VL-72B` through prompting with few-shot demonstrations. Additional bounding box metadata is used to generate both the *edit instruction* and the corresponding *thinking* field, as detailed in App. B.2.2.

### B.2.1  LLM system prompts for creating reasoning chains

> **▤  Prompt for In-context Learning of Chain-of-thought Editing**
>
> ```
> Your task is to generate a step-by-step edit plan for the given edit task based on the input image and edit instruction.
> Your reasoning should be thorough and logically structured.  Follow these guidelines:
>
> 1.  Analyze the source image in depth, describing its main elements, context, and any relevant visual details.
> 2.  Identify the object to be edited, providing specific details about its appearance, role in the scene, and
> distinguishing features.
> 3.  Clearly specify the area to be edited, including bounding box coordinates if provided, and explain why this region
> is chosen in relation to the object and the scene.
> 4.  Detail the exact changes to be made to the object or area, referencing visual attributes, position, style, and how
> the edit should be performed to maintain realism and coherence.
> 5.  Describe the expected result after the edit, focusing on how the edited image should appear, how the new or changed
> object integrates with the scene, and any requirements for preserving surrounding elements.
>
> Format your response as a concise, numbered list of steps.  Ensure each step logically follows from the previous one
> and provides sufficient depth for a clear, actionable edit plan.
> Input contains:
> - Edit Instruction:  The instruction specifying the exact edit to be made on the input image.
> - Input Image:  The source image to be edited.
> - Edited Image:  The edited image for reference (to understand how the edit instruction is applied).
> - Region Coordinates:  The region to be edited (specifies the area to be edited).
>
> Your response must be in the following JSON format:
>
> {
>         "thinking": "<Insert detailed procedural editing steps>"
> }
> ```

### B.2.2 LLM prompting for converting VisMin image-text pairs to editing task

**Prompt for In-context Learning of Chain-of-thought Editing**

Your task is to generate a step-by-step edit plan for the given edit task based on the input image and edit instruction.
Your reasoning should be thorough and logically structured.  Follow these guidelines:

1.  Analyze the source image in depth, describing its main elements, context, and any relevant visual details.
2.  Identify the object to be edited, providing specific details about its appearance, role in the scene, and
distinguishing features.
3.  Clearly specify the area to be edited, including bounding box coordinates if provided, and explain why this region
is chosen in relation to the object and the scene.
4.  Detail the exact changes to be made to the object or area, referencing visual attributes, position, style, and how
the edit should be performed to maintain realism and coherence.
5.  Describe the expected result after the edit, focusing on how the edited image should appear, how the new or changed
object integrates with the scene, and any requirements for preserving surrounding elements.

Format your response as a concise, numbered list of steps.  Ensure each step logically follows from the previous one
and provides sufficient depth for a clear, actionable edit plan.
Input contains:
- Edit Instruction:  The instruction specifying the exact edit to be made on the input image.
- Input Image:  The source image to be edited.
- Edited Image:  The edited image for reference (to understand how the edit instruction is applied).
- Region Coordinates:  The region to be edited (specifies the area to be edited).

Your response must be in the following JSON format:

{
        "edit_instruction": "<Insert edit request here>",
        "thinking": "<Insert detailed procedural editing steps>"
}

### B.2.3 Reasoning chain examples

In the case of the VisMin dataset [2], which contains two minimally changed image-text pairs, we do not receive explicit edit instructions. Instead, we infer these instructions by prompting a multi-modal large language model (MLLM) in a few-shot setting, using both the input and the edited image captions as context. The VisMin dataset covers various categories, including object, attribute, counting, and relationship. For object and attribute categories, where the task typically involves editing a specific region in the image, the target edit region (bounding box) is sufficient to generate the reasoning chain. The model infers the edit instruction by understanding the object and its properties within the defined region. In contrast, the relationship category, which describes spatial changes between objects, requires both the source image coordinates and the edited image coordinates to derive the appropriate edit instruction. For counting tasks, the reasoning chain is created by comparing the source image's bounding boxes with the removed bounding boxes in the edited image. This allows the model to infer the necessary edit to reflect the changes in object counts. We provide several examples of this process in App. B.2.3.

For other image-editing datasets, where edit instructions are explicitly provided, we prompt the MLLM with the input image, edited image, and edit instruction along with bounding box coordinates or keypoints for the objects or persons involved. This structured input allows the model to generate the reasoning chain by clearly understanding both the task and the necessary image modifications. For instance, the reasoning chain might involve identifying objects, spatially manipulating them, or removing them based on the given instructions and bounding boxes. The output reasoning chain provides a step-by-step breakdown of how the image should be edited to achieve the specified changes, ensuring that each instruction is logically followed. We also illustrate this process with further examples in App. B.2.3.

**Chain-of-thought Reasoning Examples (VisMin dataset)**

```
    {
"INPUT": {
    "input_image": "See the input image",
    "description": "remove the glasses from the face",
    "source_image_caption": "A person wearing glasses in a photo.",
    "edited_image_caption": "A person without glasses in a photo.",
    "target_edit_region": [250, 100, 350, 150]
},
"OUTPUT": {
    "thinking": "
    1. The image shows a person wearing glasses.\n\n
    2. The face is in region [250, 100, 350, 150], showing glasses.\n\n
```

```
        3. Remove the glasses while preserving natural skin texture.\n\n
        4. The output image should show a person without glasses.",
        }
}
{
    "INPUT": {
        "source_image_caption": "A bowl that is located to the left of a cat",
        "edited_image_caption": "a cat that is located to the left of a bowl",
        "source_image_coords": [
            ["a bowl", [200, 150, 300, 250]],
            ["a cat", [100, 225, 200, 275]]
        ],
        "edited_image_coords": [
            ["a bowl", [100, 225, 200, 275]],
            ["a cat", [200, 150, 300, 250]]
        ]
    },
    "OUTPUT": {
        "edit_instruction": "Move the bowl to where the cat is, and the cat to where the bowl is.",
        "thinking": "1. The image shows a bowl and a cat.
        2. To swap positions, we'll move the bowl from [200, 150, 300, 250]
        to [100, 225, 200, 275], and simultaneously relocate the cat from [100, 225, 200, 275] to [200, 150, 300, 250].
        3. This exchange will maintain each object's size and orientation while reversing their left-right relationship."
    }
}

{
    "INPUT": {
        "source_image_caption": "Two bicycles and a motorcycle.",
        "source_image_coords": [
            ["bicycle", [10, 171, 138, 256]],
            ["bicycle", [213, 171, 341, 256]],
            ["motorcycle", [331, 196, 502, 264]]
        ],
        "edited_image_caption": "One bicycle and a motorcycle.",
        "removed_bounding_boxes": [[213, 171, 341, 256]]
    },
    "OUTPUT": {
        "edit_instruction": "Remove the second bicycle.",
        "thinking": "1. The scene shows two bicycles and a motorcycle.
        2. To reduce to one bicycle, we'll remove
        the second bicycle located at [213, 171, 341, 256], leaving the first bicycle at [10, 171, 138, 256]
        and the motorcycle at [331, 196, 502, 264] untouched.
        3. This maintains proper spacing between the remaining objects."
        }
}
```

🗄 **Chain-of-thought Reasoning Examples (VisMin dataset)**

```
{
    "INPUT": {
        "edit_instruction": "Remove bicycles",
        "target_edit_region": [140, 389, 275, 431]
    },
    "OUTPUT": {
        "thinking_chain_of_thought": "Steps to remove the bicycles:
        1. The goal is to remove the bicycles from the image.
        2. Identify objects with defining bicycle
        features like wheels, handlebars, and frames.
        The bicycles are found within the bounding box [140, 389, 275, 431].
        3. Use the bounding box to isolate and erase the
        bicycles while preserving the surrounding elements.
        4. The image should appear as if the bicycles were never there.",
        }
}
```

## C   VIEScore Alignment with Human Judgment

We use the VIEScore [27] metric as our evaluation metric for the image editing task. VIEScore is based on MLLM prompting to judge the quality of a given edited image w.r.t the given editing prompt and the source image. Specifically, the MLLM is prompted twice, once to judge the semantic quality (alignment with the prompt and overediting), and perceptual quality (such as realism and artifacts). VIEScore was originally only shown to be effective on ImagenHub [28] which contains primarily object- and attribute-centric edits. We verify in Tab. 4 that correlation with human judges is also convincing on various other edit types, i.e. many of the complex edits in this paper such as spatial edits (WhatsUp column), action edits (Something Something, Action-Genome, Epic Kitchen columns) or counting (Kubric column). For this we correlate human judgements initially collected for AURORA-Bench [26] and VIEScore with a GPT4o-mini as a backbone (which we use throughout the

paper as our main metric). We note that human judges in [26] were asked to only judge the semantic alignment and not lower level visual properties such as aesthetics. Thus it is not surprising that the correlation of the semantic component of VIEScore is higher. Overall we find correlations to be on par or sometimes exceeding with those shown in the original paper [27]. For example in the original paper, the overall VIEScore (with the best model GPT4 at the time) had a correlation of 0.382 with human raters on MagicBrush. We find the lowest correlations, as expected, is on hard action-centric prompts in the Epic Kitchen subset of AURORA-Bench.

Table 4: Correlation between VIEScore (GPT4o-mini) and human judges on the 8 subtasks of AURORA-Bench [26]. Specifically, we correlate the overall VIEScore and the semantic sub-score with humans. We note that human judges in [26] were asked to only judge the semantic alignment and not lower level visual properties such as aesthetics.

| Metric | MagicBrush | Something Something | Action-Genome | Epic Kitchen | Whatsup | Kubric | CLEVR | Emu-Edit | AVG |
|---|---|---|---|---|---|---|---|---|---|
| Overall | 0.5353 | 0.3676 | 0.3585 | 0.1764 | 0.5285 | 0.2649 | 0.5600 | 0.4000 | 0.4328 |
| Semantic | 0.6300 | 0.4305 | 0.3830 | 0.2462 | 0.6011 | 0.3529 | 0.6187 | 0.5612 | 0.5288 |

# D    Analysis of the Verifier Model

## D.1    Qualitative Examples of Reward Model Limitations in Complex Edits

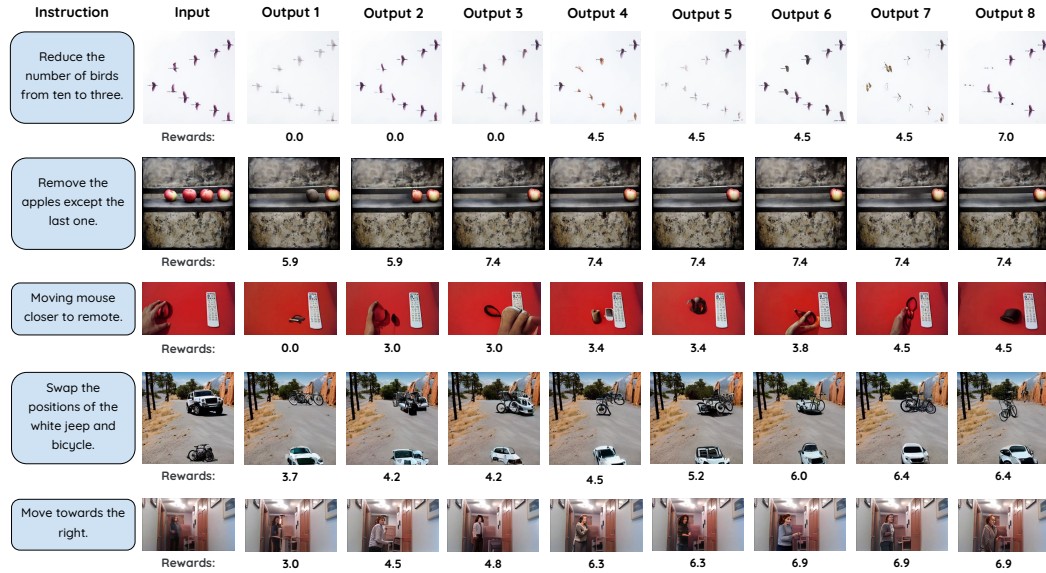

Figure 6: Examples of limitations of the verifier for the complex edits. The reward values correspond to the VIEScore given by our verifier.

Our reward model shows strong performance on simple edits involving object and attribute changes but exhibits limitations when verifying complex edits, as illustrated by several qualitative examples in Fig. 6.

In **Row 1**, the task is to reduce the number of birds from 10 to 3. This large-count change is difficult for the verifier to handle accurately, reflected in inconsistent scores across different outputs. For instance, although output 2 and 6 are quite similar, the rewards they receive differ significantly (0.0 vs. 4.5). This indicates uncertainty in reliably evaluating edits that involve large numerical changes. In contrast, **Row 2** is an example of small count changes. It requires removing all apples except one. The verifier assigns consistently high scores to the correct generations (outputs 3 to 8), suggesting that smaller count changes are easier to verify, likely because the verifier is more exposed to such data during training.

In **Row 3**, the task involves moving a mouse closer to a remote. Although the model generally assigns higher scores to better samples, it struggles with inconsistencies: in output 8, the mouse's appearance changes noticeably, and in output 7, the hand shows some distortion. This underscores the difficulty of verifying subtle positional changes compared to simpler edits, such as additions and removals.

For **Row 4**, the edit requires swapping the positions of a white jeep and a bicycle. The verifier correctly favors outputs that accurately reflect this positional change and exhibit fewer artifacts and distortions, demonstrating sensitivity to spatial relationships and artifacts. For example, outputs 2 and 3 still show traces of the car in its original position, indicating both leftover artifacts and a failure to correctly reflect the position swap. However, obtaining fully reliable reward signals for fine-grained spatial relationships still remains challenging.

Finally, **Row 5** involves moving a person toward the right side. Although the verifier prefers outputs where the person's position is more towards the right, it does not take into account the changes in person's appearance.

It is important to note that while verifier scores vary, the model can still provide meaningful signals for some complex edits such as detecting lower object counts or changes in position to a certain extent, helping guide improvements in edit quality.

### D.2  VIEScore Evaluation with Qwen2.5-VL-72B and Alignment with Human Judgment

We also report the correlation between Qwen2.5-VL-72B's judgments (as captured by VIEScore) and human judgments on AURORA-Bench (columns show results for individual subtasks, and "AVERAGE" denotes the aggregate performance). While the absolute correlation values are moderate (around 0.4), they are consistent with those reported in the original VIEScore paper (0.382) [27], which established VIEScore as a reliable automatic evaluation metric. This consistency supports the reliability of Qwen2.5-VL-72B as a verifier. Moreover, the results indicate that Qwen2.5-VL-72B performs robustly not only on object- and attribute-centric edits but also on more complex edits such as spatial (WhatsUp) and action (Something-Something, Action-Genome).

Table 5: Correlation results between Qwen judgments and human judgments across different datasets. Although the absolute values are modest, they are comparable to the established VIEScore benchmark.

| **Metric** | MagicBrush | Something Something | Action-Genome | Epic Kitchen | Whatsup | AVERAGE |
|---|---|---|---|---|---|---|
| Overall | 0.5377 | 0.3084 | 0.4385 | 0.2393 | 0.4420 | 0.3971 |
| Semantic | 0.6104 | 0.3192 | 0.4392 | 0.2886 | 0.4556 | 0.4309 |

### D.3  Impact of Verifier Choice on RL Performance

Our main experiments employ Qwen2.5-VL-72B as the reward model (verifier) for RL post-training. We selected Qwen2.5-VL-72B because of its reliable reward feedback and strong performance on fine-grained vision-language tasks, comparable to GPT-4o. We also explored alternative verifiers to study their effect on reward signal quality and overall training performance. Notably, small-sized multimodal LLMs often produced noisy outputs, particularly in tasks requiring *instruction-following* and *prompt adherence*. We tested an instruction-tuned model, Qwen-7B. As shown in Tab. 6, replacing Qwen2.5-VL-72B with Qwen-7B led to a significant drop in performance: from an average score of **3.88** with SFT alone to **2.81** with RL post-training. This highlights that smaller models cannot provide sufficiently accurate or stable feedback for AR-based RL training, underscoring the importance of a strong verifier.

Table 6: Impact of verifier choice on model performance (higher is better).

| **Method** | AURORA | MagicBrush | OmniEdit | I2EBench | Vismin | EmuEdit | **Average** |
|---|---|---|---|---|---|---|---|
| SFT (S) | 3.58 | 3.19 | 5.73 | 3.59 | 3.57 | 3.66 | **3.88** |
| SFT (S) → RL (S+C) [Qwen-7B] | 2.55 | 2.66 | 4.05 | 2.60 | 2.32 | 2.69 | **2.81** |

# E Analysis of Model Outputs

## E.1 Breakdown of VIEScore

To better quantify model behavior, we show four components of VIEScore: Edit Success and OverEdit, which reflect semantic alignment with editing instructions, and Looks Natural and No Artifact, which capture perceptual quality. Tab. 7 reports these metrics for both the SFT(S) baseline and our best-performing configuration, SFT(S) → RL(S+C). As shown in the AVERAGE column, RL notably improves Edit Success, demonstrating stronger alignment with user instructions, while maintaining comparable performance on OverEdit and preserving perceptual quality in terms of Looks Natural and No Artifact.

Table 7: Breakdown of VIEScore into four components: Edit Success, OverEdit, Looks Natural, and No Artifact comparing SFT(S) and SFT(S) → RL(S+C) across six benchmarks. Values are reported as SFT(S) / SFT(S) → RL(S+C), with the higher score shown in bold.

| Metric | OmniEdit | EmuEdit | AURORA | MagicBrush | VisMin | I2EBench | AVERAGE |
|---|---|---|---|---|---|---|---|
| Edit Success | 6.1 / **7.0** | 3.5 / **4.2** | 3.0 / **3.6** | 2.6 / **4.1** | 3.2 / **4.3** | 3.7 / **4.3** | 3.7 / **4.6** |
| OverEdit | 7.3 / **7.5** | **7.5** / 7.0 | **7.0** / 6.6 | **7.8** / 7.0 | **5.7** / 6.5 | 6.8 / **6.9** | **7.0** / 6.9 |
| Looks Natural | 6.7 / **6.9** | **6.3** / 6.0 | **5.8** / 5.7 | **6.3** / 5.8 | 6.6 / **6.8** | 5.0 / **5.1** | **6.1** / 6.0 |
| No Artifact | 7.5 / **7.7** | **6.9** / 6.7 | 6.8 / **6.7** | **6.9** / 6.4 | 7.3 / **7.6** | 5.8 / **5.9** | **6.9** / 6.8 |
| VIEScore | 5.7 / **6.3** | 3.7 / **4.3** | 3.6 / **4.0** | 3.2 / **4.3** | 3.6 / **4.5** | 3.6 / **4.1** | 3.9 / **4.6** |

## E.2 CLIP Similarity and FID Evaluation

To provide a more comprehensive and fair evaluation beyond VIEScore, we further report two widely used metrics: CLIP similarity and FID (Fréchet Inception Distance). These metrics capture complementary aspects of model performance: CLIP similarity measures the semantic consistency between generated and ground-truth edited images (↑ higher is better), while FID quantifies image realism and distributional closeness (↓ lower is better). We observe that EARL achieves competitive performance on both CLIP similarity and FID, remaining within the same range as strong baselines such as Omnigen, Aurora, and EditAR. This demonstrates that EARL preserves semantic alignment and image quality while providing comparable fidelity to existing methods. Results are averaged across three benchmarks: MB, VisMin, and OmniEdit.

Table 8: **CLIP Similarity (ViT-B/32) – Edited Image vs. Ground Truth.** Higher scores indicate stronger semantic alignment.

| Method | MB | InstructPix2Pix | Aurora | Omnigen | EditAR | EARL SFT(S)→RL(S+C) |
|---|---|---|---|---|---|---|
| CLIP | 0.91 | 0.85 | 0.89 | 0.90 | 0.88 | 0.88 |

Table 9: **FID (InceptionV3) – Edited Images vs. Ground Truth.** Lower scores indicate higher image realism.

| Method | MB | InstructPix2Pix | Aurora | Omnigen | EditAR | EARL SFT(S)→RL(S+C) |
|---|---|---|---|---|---|---|
| FID | 40.59 | 53.20 | 47.22 | 45.22 | 49.47 | 49.64 |

## E.3 Additional Comparison with EditAR

As EditAR [38] is the only existing autoregressive baseline for image editing, we conduct a more comprehensive comparison with it. Our best-performing setup (SFT(S) → RL(S+C)) achieves an average VIEScore of 4.57, surpassing EditAR's 4.20 across all benchmarks. EditAR also reports results on PIEBench [24] for image editing. As shown in Tab. 10, EARL outperforms EditAR on five metrics, including Structure Distance score, PSNR, LPIPS [62], MSE, and SIM [51]. On CLIP similarity scores computed using ViT-Large-14 on the whole image (CLIP-W) and on the edited regions (CLIP-E), EARL slightly underperforms compared to EditAR. However, our scores remain comparable to EditAR, demonstrating strong perceptual quality and better background preservation.

Table 10: PIEBench results comparing EARL with EditAR (most similar work to ours). SD refers to Structure Distance and CLIP-W, CLIP-E refers to CLIP similarity scores on the whole image and edited regions, respectively.

| Method | Base Model | SD ↓ | PSNR ↑ | LPIPS ↓ | MSE ↓ | SSIM ↑ | CLIP-W ↑ | CLIP-E ↑ |
|--------|-----------|------|--------|---------|-------|--------|----------|----------|
| EditAR | LlamaGen | 39.43 | 21.32 | 117.15 | 130.27 | 75.13 | **24.87** | **21.87** |
| EARL | Emu3 | **35.00** | **23.71** | **112.40** | **77.00** | **79.17** | 24.44 | 21.34 |

Table 11: Best-of-N evaluation results for SFT and RL-tuned models across six benchmarks

| Model | OmniEdit | EmuEdit | AURORA | MB | VisMin | I2EBench | AVG |
|-------|----------|---------|--------|------|--------|----------|------|
| SFT (S) | 5.73 | 3.66 | 3.58 | 3.19 | 3.57 | 3.59 | 3.88 |
| SFT (S) [Best of 5] | 6.13 | 4.21 | 3.89 | 3.63 | 4.09 | 4.00 | 4.33 |
| SFT (S) → RL (S+C) | 6.33 | 4.28 | 3.99 | 4.26 | 4.48 | 4.08 | 4.57 |
| SFT (S) → RL (S+C) [Best of 5] | 6.54 | 4.66 | 4.08 | 4.66 | 4.69 | 4.36 | 4.83 |

### E.4 Best-of-N Evaluation

We further evaluate model performance under a best-of-N sampling regime to separate the effects of sampling diversity from policy optimization. As shown in Tab. 11, best-of-5 sampling substantially improves SFT results (3.88 → 4.33), indicating significant headroom through better candidate selection. However, the RL-tuned model still outperforms SFT even when both use best-of-5 (4.33 → 4.83), demonstrating that policy optimization contributes improvements beyond what sampling alone can achieve. Moreover, the small gap between RL single-sample and RL best-of-5 (4.57 → 4.83) suggests that the RL policy is already producing high-quality outputs with minimal reliance on sampling, highlighting the effectiveness of our training approach.

### E.5 Fine-Grained Evaluation of EARL

**Qualitative** Fig. 7 shows EARL SFT (S) → RL (S+C) performing four types of image edits: counting, action, spatial, and simple; each illustrated by three side-by-side input/output examples. In the counting row, EARL correctly removes one poodle and two toy cars but fails to remove one egg from the third image. In the action row, it successfully takes the white cup out and opens the orange bag in the first two examples, but cannot make the person stand up in the final example. For spatial edits, the model effectively interprets spatial relationships by correctly removing the left fire hydrant and adding a man to the left of the road sign, but it fails to add a picture to the left of the woman in the third example. Finally, in the simple edits row, EARL recolors an alien spaceship pink and successfully erases palm tree with clean backgrounds, but fails to edit the third image and removes the bowling bowl instead of the truck. Overall, these examples highlight the model's strong potential in performing complex image edits, while also indicating some challenges that remain to be addressed.

**Quantative** The quantitative results ( Tab. 12 13 14 15 16 17) demonstrate that EARL SFT (S) → RL (S+C) consistently outperforms the baseline SFT (S) across multiple fine-grained editing tasks and datasets. For instance, in Tab. 13 on I2EBench, improvements are evident in counting, direction perception, object manipulation, and style alteration metrics. Similarly, Tab. 14 shows gains in style, attribute modification, environment changes, and object addition/removal on OmniEdit. Tab. 15 further confirms these trends on EmuEdit, with better performance in object addition, background editing, and both global and local changes. Lastly, Tab. 16 highlights improved counting and spatial relation accuracy on VisMin. On AURORA, as shown in Tab. 17, there is no improvement in action edits for EARL SFT (S) → RL (S+C) when compared to SFT (S), mainly because SFT (S) itself was very poor on action edits. Overall, these results underscore the effectiveness of combining SFT with RL for diverse kinds of edit types.

### E.6 Qualitative Evaluation of Reasoning and Edits in the SFT-Think Model

Fig. 8 presents qualitative examples comparing two model variants: SFT(S), and SFT think(S).

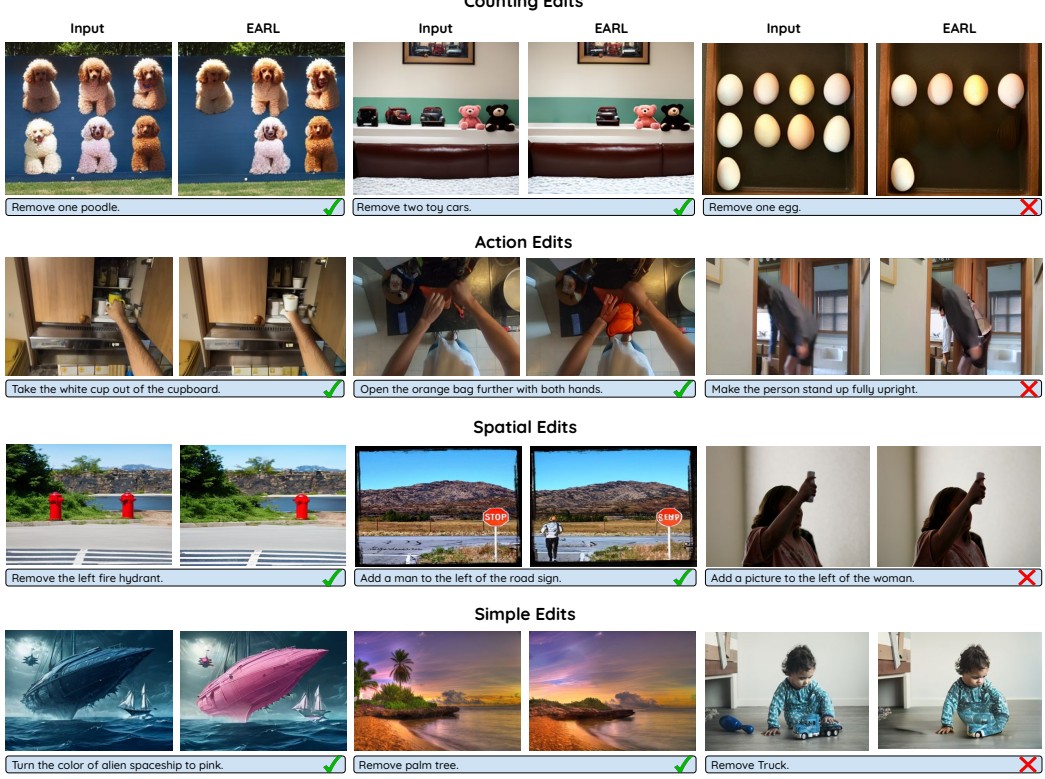

Figure 7: Qualitative examples of EARL across diverse edit types—counting, action, spatial, and simple.

Table 12: Fine-grained evaluation of SFT and RL model variants on I2EBench (Low-level Editing).

| Model/Edit Task Category | Deblurring | Haze Removal | Lowlight Enhancement | Noise Removal | Rain Removal | Shadow Removal | Snow Removal | Watermark Removal |
|---|---|---|---|---|---|---|---|---|
| SFT (S) | 2.36 | 3.73 | 2.56 | 2.36 | 3.98 | 4.60 | 2.59 | 3.82 |
| EARL SFT (S) → RL (S+C) | 2.33 | 3.51 | 3.03 | 3.04 | 4.74 | 4.98 | 3.16 | 4.28 |

Table 13: Fine-grained evaluation of SFT and RL model variants on I2EBench (High-level Editing).

| Model/Edit Task Category | Counting | Direction Perception | Object Removal | Object Replacement | Background Replacement | Color Alteration | Style Alteration | Region Accuracy |
|---|---|---|---|---|---|---|---|---|
| SFT (S) | 2.74 | 1.82 | 4.52 | 2.49 | 3.47 | 5.74 | 5.34 | 4.36 |
| EARL SFT (S) → RL (S+C) | 4.08 | 3.45 | 4.76 | 3.40 | 4.72 | 5.98 | 5.80 | 5.06 |

Table 14: Fine-grained evaluation of SFT and RL model variants on OmniEdit.

| Model/Edit Task Category | Style | Attr. Mod. | Env | Swap | Addition | Removal |
|---|---|---|---|---|---|---|
| SFT (S) | 5.52 | 6.30 | 6.12 | 5.69 | 4.86 | 5.91 |
| EARL SFT (S) → RL (S+C) | 5.99 | 6.95 | 6.68 | 6.39 | 6.00 | 6.30 |

Table 15: Fine-grained evaluation of SFT and RL model variants on EmuEdit.

| Model/Edit Task Category | Add | Remove | Background | Text | Color | Style | Global | Local |
|---|---|---|---|---|---|---|---|---|
| SFT (S) | 2.36 | 4.91 | 2.47 | 2.15 | 4.91 | 4.89 | 4.14 | 3.78 |
| EARL SFT (S) → RL (S+C) | 4.46 | 4.81 | 3.24 | 3.16 | 5.85 | 5.13 | 4.77 | 4.47 |

**Example 1** This example involves changing the color of a wooden rabbit sculpture to brown, while preserving its carved details and natural background.

Table 16: Fine-grained evaluation of SFT and RL model variants on VisMin.

| Model/Edit Task Category | Counting | Spatial Relation |
|---|---|---|
| SFT (S) | 4.22 | 2.91 |
| EARL SFT (S) $\rightarrow$ RL (S+C) | 5.72 | 4.14 |

Table 17: Fine-grained evaluation of SFT and RL model variants on AURORA. $^*$Action Genome subset of the Aurora benchmark specifically contains complex action edits.

| Edit Task | SFT (S) | EARL SFT (S) $\rightarrow$ RL (S+C) |
|---|---|---|
| $^*$Action | 3.09 | 2.96 |

- SFT (S): Performs a successful edit without reasoning, changing the color to brown while preserving the background unchanged.
- SFT think (S): Uses Chain-of-Thought (CoT) reasoning, including scene description, object identification, and bounding box localization. It understands the edit instructions and grounds the plan well. However, the edited rabbit appears unnatural and loses some details.
- **Observation:** Although SFT think(S) demonstrates strong reasoning and grounding capabilities, it introduces artifacts or unnatural features in the edited image compared to the simpler SFT(S) variant.

**Example 2** This example involves replacing an antique wooden radio with a typewriter, maintaining the spatial layout and removing the original radio's elements.

- SFT (S): Successfully replaces the radio with the typewriter, keeping the proportions and orientation consistent.
- SFT think (S): Uses Chain-of-Thought (CoT) reasoning to carefully plan the replacement, including brand details, but shows minor inconsistencies by mentioning two different brand names (IBM and Makoka). The final edited image presents slight visual distortions around the edges and keys.
- **Observation:** While the reasoning model provides detailed planning and grounding, it sometimes produces inconsistent details—such as mentioning two different brand names (IBM and Makoka)—reflecting a pattern of hallucination in the reasoning process that can reduce visual quality compared to the non-reasoning model.

**Example 3** This example involves transforming a green forested area around a small building into a garden with blooming flowers while keeping the building unchanged.

- SFT (S): Successfully changes the environment to a blooming garden without reasoning, preserving the building and overall composition.
- SFT think (S): Uses Chain-of-Thought (CoT) reasoning to plan the seasonal transformation carefully, resulting in a well-grounded and detailed edit.
- **Observation:** Both models perform the edit well, delivering accurate and natural-looking results.

In summary, the reasoning model can ground and plan edits, but it may sometimes reduce visual quality by introducing artifacts and unnatural details compared to the no-reasoning model.

# F   Experimental Setup and Compute Resources

## F.1   Training Details

For SFT training, we used the AdamW optimizer with weight decay 0.1 and parameters $\beta_1 = 0.9$, $\beta_2 = 0.95$, and $\epsilon = 1 \times 10^{-6}$. The learning rate followed a cosine schedule with a minimum value of $10^{-6}$, and gradient clipping was applied with a maximum norm of 5.0. Training employed

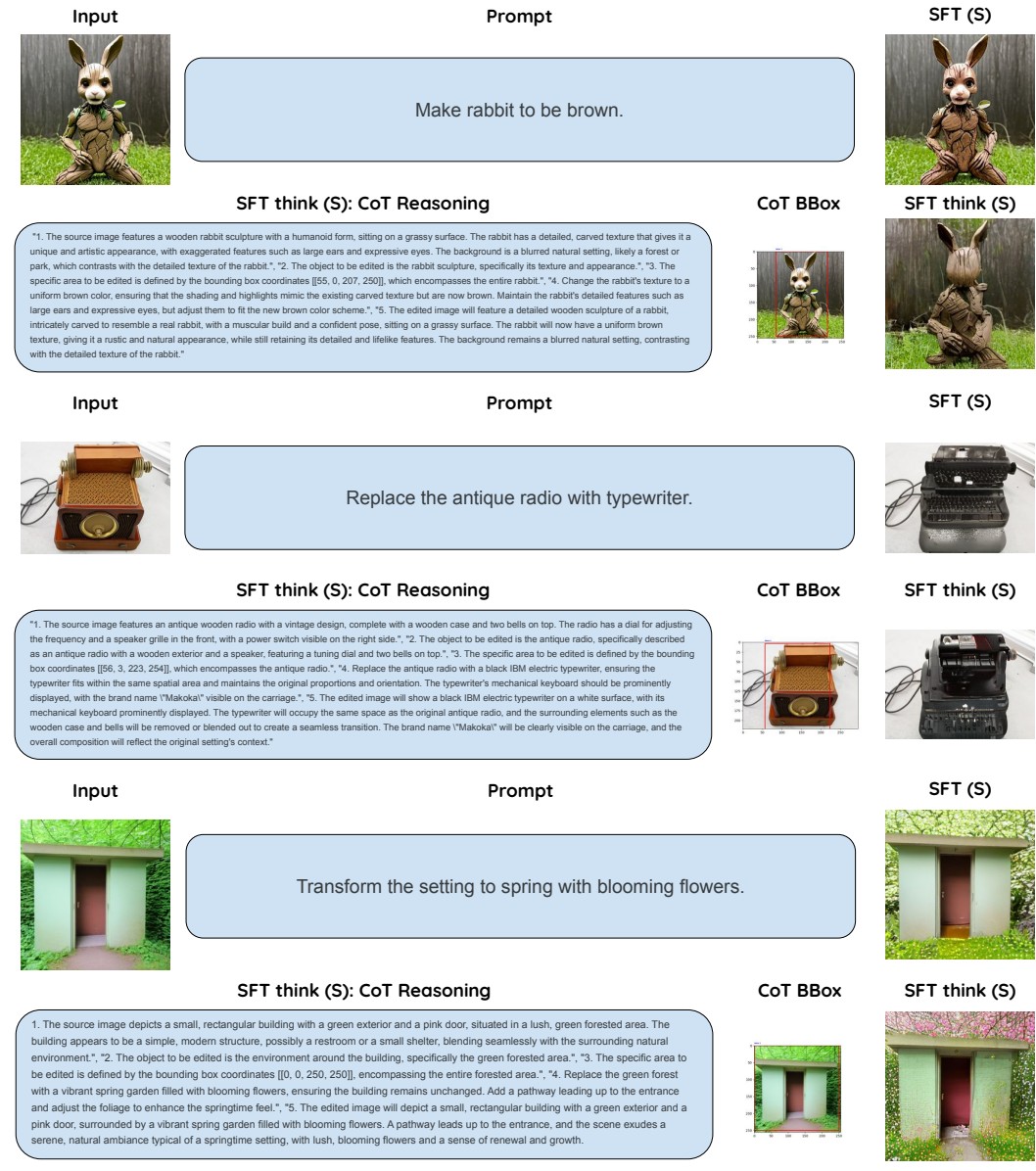

Figure 8: This figure shows image editing examples from SFT(S) without reasoning, SFT think(S) with Chain-of-Thought (CoT) reasoning. While the reasoning model can understand instructions, plan, and ground edits, it may introduce artifacts and unnatural details compared to the non-reasoning model.

DeepSpeed ZeRO stage 3 for memory efficiency, with mixed precision enabled using bfloat16 (bf16). We train the model for a maximum of 5 epochs.

For RL training, rewards were computed using VIEScore through a vLLM API server running Qwen2.5-VL-72Bon 4 NVIDIA H100 GPUs. The training was conducted separately on a different server for 2000 steps, with early stopping based on reward plateaus, also using four NVIDIA H100 GPUs.

## F.2 Training Efficiency Comparison

We compare the training data size, compute resources, and training duration of EARL with existing baselines, as shown in Tab. 18. EARL achieves competitive performance with roughly $5\times$ less image-editing data (752k vs. 4M samples) and significantly fewer GPUs (8 vs. 104) compared to Omnigen [56], demonstrating superior data and training efficiency. Note that the higher GPU demand for Omnigen is expected, as its model is trained from scratch. In contrast, EARL builds on pretrained models, reducing computational requirements. This highlights that EARL offers practical efficiency gains while maintaining strong performance.

Table 18: Comparison of training data size, compute resources, and training duration for **EARL** and prior methods. "–" indicates values not reported.

| Model | Training Data | Compute | Duration |
|---|---|---|---|
| **EARL SFT(S) → RL(S+C)** | ~752k samples (SFT: ~750k, RL: 32k) | 8×A100L | 108h (SFT: ~60h, RL: ~48h) |
| InstructPix2Pix [8] | 313k samples | 8×A100 | 25.5h |
| MagicBrush [61] | ~10k samples (built on Instruct-Pix2Pix) | 2×A100 | – |
| Aurora [26] | 289k samples (built on Instruct-Pix2Pix) | 2×RTX A6000 | 16h |
| Omnigen [56] | ~0.1B samples (incl. ~4M editing examples) | 104×A800 | – |
| EditAR [38] | 1.5M samples | 8×A100 | – |

## F.3 Inference Speed Comparison

We integrated EARL with vLLM [29], which enables fast autoregressive decoding through PagedAttention and optimized parallel token generation, significantly reducing latency and improving throughput compared to standard autoregressive decoding. We evaluated inference speed on 50 samples using a single A100L GPU, running all models at a resolution of $256 \times 256$ (except EditAR, which was trained and tested at $512 \times 512$). The total times to generate 50 samples were as follows: EARL: 52.7 s, EditAR: 66.19 s, MagicBrush: 23.6 s, InstructPix2Pix: 23.7 s, Aurora: 23.7 s, and Omnigen: 200 s. Compared to Omnigen, EARL is roughly **4× faster** while achieving competitive editing performance (see Table 2), demonstrating the benefits of optimized autoregressive generation. Although EARL is about 2× **slower** than diffusion-based baselines such as MagicBrush, InstructPix2Pix, and Aurora, this trade-off is justified by its significant improvements in editing quality.

# G Limitations and Broader Impact

## G.1 Limitations

First, the model's performance heavily depends on the coverage of our training data. Although we curated a diverse set of simple and complex edit triplets, long-tail concepts such as fine-grained cultural artifacts, specialized scientific diagrams, and underrepresented geographic scenes remain sparsely represented. This limited coverage can lead to brittle behavior when the model encounters out-of-distribution inputs.

Second, our reinforcement learning approach relies on a single frozen vision-language verifier, which, despite being a state-of-the-art MLLM, has inherent limitations. The verifier can be imperfect and it inherits biases from its pretraining corpus. It struggles particularly with verifying complex edit types involving spatial relationships and action changes. Although these edits are underrepresented in the verifier's training data, qualitative and quantitative analyses indicate that the verifier still provides meaningful learning signals to guide the model.

Lastly, our training data depends on synthetic data generated via diffusion models, which include automatic filtering to reduce noise. However, some noisy examples remain, such as image deformations or outputs that do not accurately reflect the edit instructions, due to imperfections in synthetic data generation. These factors can introduce noise in training. However, some of these can be improved during RL post-training, as RL post-training does not need the ground-truth labels for edited images.

### G.2 Broader Impact

**Positive Impacts**   Text-guided image editing systems such as our RL-enhanced approach has the potential to amplify human creativity and promote accessibility. By accepting natural language instructions, the model helps lower the barrier for designers, educators, and hobbyists who lack advanced editing expertise, enabling rapid iteration on visual concepts. Beyond direct applications, our RL pipeline offers a way to overcome the requirement of ground-truth edited images.

**Potential Negative Impacts**   While high-fidelity image editing offers many benefits, it also poses risks. Such technology can be misused to create convincing misinformation or deepfakes. Additionally, if the base model or the vision-language verifier contains demographic biases, reinforcement learning may inadvertently amplify these biases. Importantly, our model is developed strictly for research purposes and is intended to advance scientific understanding rather than for deployment in real-world applications. We encourage ongoing efforts to implement safeguards and promote ethical use alongside further development in this area.

## H   Behind the Scenes

We started this project with a clear goal: to build a single, unified model that edits images guided only by text instruction – no user-provided masks, bounding boxes, or conditions. The model itself would reason and plan the edits, generating all necessary guidance on its own. The rationale behind this is that diffusion models – a dominant approach to image editing are typically built with some form of user conditioning to control for faithful editing, and they lack unification – a separate model built for different edit types. We wanted to have a unified model that does all sorts of edits and treats user conditioning as a learnable task via reasoning. To achieve this, we needed a model capable of generating both images and text in one combined sequence, so we searched for an interleaved image-text transformer model.

The first model we found is Meta's Chameleon model [45], as it fits the criteria well. It had good image generation capabilities, but the model was not open source. So, we pivoted to Anole [9], an open-source variant inspired by Chameleon. We trained Anole to do image editing without any reasoning input (the sanity check one can do). Unfortunately, the results were rough: images suffered from distortion and artifacts, making it clear we needed something stronger.

Then came Emu3 [50]. When it was released, we quickly tested it, fine-tuning with small, complex editing datasets. Results initially fell short – until we added large amounts of simpler, high-quality edit data. This mix showed improvements: the model began handling simplistic edits well; also, some hope for counting, spatial, and even some action edits successfully.

Our main goal is to integrate reasoning to eliminate the need for user-provided conditioning. To this end, we introduced reasoning data – structured "chain of thought" prompts to guide the model's editing process, as detailed extensively in our paper. Surprisingly, models trained with reasoning mostly underperformed compared to those without it. We tried many variants: changing the data pool, simplifying edit tasks, shortening the chain-of-thought length, and varying training data size, but none made a significant difference. We suspect this gap is due to the base model's capacity or the quality of the reasoning data. Even feeding ground-truth reasoning directly as input during fine-tuning failed to boost results as much as expected.

One positive result we had with the no-reasoning variant is that we were able to get close to state-of-the-art. Since RL has been gaining traction lately, we were eager to explore its potential by applying it to our problem, but we needed to build a strong SFT model first. We tested SFT models with generating multiple samples given a fixed prompt. One interesting observation was that at least one of these samples did well on simple edit tasks (e.g. changing object, attribute), and sometimes even a complex one (e.g. changing a count). That flicker of success motivated us to integrate RL on top of SFT-ed models – to push the model towards consistently generating better edits. Compared to our exploration of teaching an autoregressive model editing through reasoning, which was for most part unsuccessful for a long while, RL experiments were more promising early on – clearly moving our interest to go further with the RL route.

This journey has been a rollercoaster of setbacks and great excitement when RL was working consistently. Yet, each experiment deepened our understanding of the delicate balance between

data quality, reasoning guidance and RL – bringing us closer to that elusive goal of truly unified, instruction-guided image editing.

# I  Author Contributions

Saba Ahmadi, Rabiul Awal, and Benno Krojer initiated the project. Saba Ahmadi led the design and implementation of the EARL for auto-regressive image editing. Saba and Rabiul initiated the idea of incorporating reasoning into image editing. Rabiul and Ankur Sikarwar led the data generation for the CoT reasoning experiments. Ankur also led the evaluation. Amirhossein Kazemnejad led the design and the implementation of the reinforcement learning pipeline. Ge Ya Luo and Benno Krojer helped with evaluation metric selection. Rabiul, Saba, Ankur, Amirhossein, and Juan A. Rodriguez ran experiments for different stages of the project. Benno wrote the introduction and provided guidance on individual sections. Saba, Rabiul, Ankur, and Amirhossein led the writing of the remaining sections with feedback from the PIs. Aishwarya Agrawal, the lead PI, guided the project from the start, with additional guidance from Siva Reddy, Christopher Pal, and Sai Rajeswar at various stages.

