# OpenReview forum: "The Promise of RL for Autoregressive Image Editing"
_NeurIPS.cc/2025/Conference — NeurIPS 2025 poster_

### Official Review · Reviewer_WQB5 · 2025-06-28

**Clarity:** 3
**Significance:** 4
**Originality:** 3
**Rating:** 5
**Confidence:** 4

**Summary:**

Authors propose EARL, a strong autoregressive image editing method trained with GRPO. EARL besta other autoregressive editing methods and matches the performance of strong diffusion based editing methods.

**Questions:**

-

**Ethical Concerns:**

["NO or VERY MINOR ethics concerns only"]

**Final Justification:**

Good paper with decent motivation and experiments.

I'll keep my score.

**Limitations:**

yes

**Quality:**

3

**Strengths And Weaknesses:**

Strengths:
- Shows a supervised finetuning, reinforcement learning, and CoT reasoning for image editing.
- Achieves the best AR editing performance matching Diffusion editing performance.
- Good qualitative analysis on types of images selected and rejected in the GRPO.

Weaknesses:
- EditAR results are only shown with PIE Bench. There is need for evaluating EditAR on the general and complex image benchmarks tested in Table 2 - OmniEdit, EmuEdit, etc.
- There is no evidence that COT works for image editing and yet the authors spend a significant portion of the paper discussing that.
- Why wasn't the training done with more data similar to OmniGen?
- Only GRPO explored; PPO/LORA-PPO comparisons would strengthen generality of strength of RL.
- No clear reasoning as to why SFT (S) → RL (S+C) is the best combination. Why are some of the other SFT(.) → RL (.) combinations not as good.
- Some discussion on how would one extend this method to diffusion image editing would be helpful.

---

> ### Author Rebuttal · Authors · 2025-07-31
>
> Thank you for the positive feedback. We appreciate your recognition of our **wide study of supervised finetuning, reinforcement learning, and CoT reasoning for image editing**. We are glad you recognized that **our model yields the best AR editing performance**, **competitive with diffusion methods**, and appreciated the **qualitative analysis of GRPO-selected and rejected samples**.
>
> >EditAR results are only shown with PIE Bench. There is need for evaluating EditAR on the general and complex image benchmarks tested in Table 2 - OmniEdit, EmuEdit, etc.
>
> As mentioned in Section 5.2, EditAR was not open-sourced at the time of submission, so we reported its results only on PIEBench (which is the only benchmark on which EditAR reports results for image editing). However, the code is now available, and we have evaluated EditAR on all the benchmarks used in our study, including OmniEdit, EmuEdit, AURORA, MagicBrush, VisMin, and I2EBench. In our best-performing setting (SFT(S) → RL(S+C)), EARL outperforms EditAR  on the AVERAGE metric, further confirming the effectiveness of our RL-based autoregressive editing approach. We will include these results in the final version.
>
>
> | Baseline              | AURORA | MagicBrush | OmniEdit | I2EBench | VisMin | EmuEdit | AVERAGE |
> |------------------------|--------|------------|----------|----------|--------|---------|---------|
> | EditAR                | 3.79   | 3.84       | 5.29     | 3.84     | 4.54   | 3.88    | 4.20    |
> | EARL SFT(S)→RL(S+C)   | 3.99   | 4.26       | 6.33     | 4.08     | 4.48   | 4.28    | 4.57    |
>
> > There is no evidence that COT works for image editing and yet the authors spend a significant portion of the paper discussing that.
>
> Our goal was to systematically evaluate the three training paradigms: SFT, RL, and SFT with CoT reasoning. This setup provides a comprehensive understanding of their impact on AR image editing. We included CoT because it has shown promise in NLP reasoning tasks. So, even though CoT did not lead to performance improvements,  we believe our findings will **inform** future work in image editing.
>
>
> We also note that Reviewer 1 finds this comparative analysis valuable, and specifically highlights that CoT’s limited effectiveness in our experiments  “underscores the clear motivation for integrating autoregressive generation with RL.” Including CoT thus contributed to a more complete and grounded evaluation of training strategies in this setting.
>
> >Why wasn't the training done with more data similar to OmniGen?
>
> Omnigen uses ~5× more image editing data (752k image editing sample for EARL vs 3.7M image editing samples for Omnigen) but as an academic lab, we operate under limited computational resources. Our focus was on systematically comparing training paradigms (SFT, RL, CoT) rather than scaling data. That said, our method is fully scalable, and users with more resources can readily extend it with larger datasets.
>
> **`EARL Scaled RL Post‑Training`** To demonstrate the scalability of our method,  we scaled up the RL training of our the best-performing setup (SFT(S) → RL(S+C)) to 2000 steps with 32k samples (vs. 1000 steps and 1.6k samples used in the review version), The resulting model, EARL, achieves strong performance compared to Omnigen  on AVG (as shown in the following table),—demonstrating the scalability of our approach.
>
> | Model   | OmniEdit | EmuEdit | AURORA | MagicBrush | VisMin | I2EBench | AVG  |
> |---------|----------|---------|--------|------------|--------|----------|------|
> | EARL    | **6.39** | 4.47    | **4.27** | 4.52       | **4.93** | 4.19     | **4.80** |
> | Omnigen | 5.68     | **5.00** | 4.10   | **4.68**   | 4.09   | **4.68** | 4.70 |
>
> >Only GRPO explored; PPO/LORA-PPO comparisons would strengthen generality of strength of RL.
>
> We focused on GRPO due to its strong stability and efficiency in large-scale autoregressive training compared to methods like PPO, as demonstrated in recent LLM literature [1]. While comparing with PPO or LoRA-PPO could further validate the generality of our approach, our primary goal was to establish the viability of RL for autoregressive image editing and to provide the community with a strong RL-based AR model that can be further built on top of.
>
> [1] DeepSeek-R1: Incentivizing Reasoning Capability in LLMs via Reinforcement Learning. arXiv:2501.12948.
>
> >No clear reasoning as to why SFT (S) → RL (S+C) is the best combination. Why are some of the other SFT(.) → RL (.) combinations not as good.
>
> We found that **SFT(S)** produces the strongest and the most stable base model. This is because high quality paired data (required for SFT) is easily available for simple edits, allowing the model to reliably learn the basic editing task. In contrast, **SFT(S+C)** underperforms due to the limited size, diversity, and quality of the complex edit data (e.g., spatial reasoning, actions), which results in more noisy  learning.
>
> For the RL stage, **training on both simple and complex edits (S+C)** proves most effective. RL on only complex edits **(RL(C))** suffers from a strong distribution shift relative to the base model and often fails to produce usable outputs, making it hard for GRPO training to receive meaningful learning signals. RL on only simple edits **(RL(S))** is stable but lacks coverage for harder tasks. **RL(S+C)** balances stability and coverage: simple edits provide reliable gradients, while complex edits gradually extend the model’s capabilities.
>
> > Some discussion on how would one extend this method to diffusion image editing would be helpful.
>
> Extending our verifier-based RL setup to diffusion models is a promising direction. Since our verifier operates on the final output image, the overall setup is, in principle, applicable to both diffusion and autoregressive models. The key challenge lies in adapting the GRPO algorithm for the diffusion modeling family. However, recent work such as Flow-GRPO [2] demonstrates that adapting policy gradient methods like GRPO is feasible for diffusion model families. While this remains an active area of research and is beyond the scope of our study, our strong results underscore that autoregressive models already offer a simple, effective, and practical solution for image editing, thus establishing autoregressive modeling as a strong contender to diffusion modeling for image editing.
>
> [2] Flow-grpo: Training flow matching models via online RL. arXiv preprint arXiv:2505.05470.

---

> > ### Comment · Reviewer_WQB5 · 2025-08-03
> > **Response**
> >
> > Thank you for the additional results with EditAR and clarifications. I've already given a good score and will keep it.

---

> > > ### Author Response · Authors · 2025-08-05
> > > **Reply to Reviewer WQB5**
> > >
> > > We sincerely thank the reviewer for the positive feedback and support.

---

### Official Review · Reviewer_vNkF · 2025-06-28

**Clarity:** 3
**Significance:** 3
**Originality:** 3
**Rating:** 4
**Confidence:** 3

**Summary:**

This work investigates three training paradigms—supervised fine‑tuning (SFT), reinforcement learning (RL), and chain‑of‑thought (CoT) reasoning—for a unified autoregressive multimodal model (Emu3) on text‑guided image editing. The authors propose EARL, which applies GRPO‑based RL post‑training with a large MLLM verifier (Qwen2.5‑VL‑72B) and demonstrate that this approach yields significant gains over SFT alone, achieving competitive or superior performance to strong diffusion‑based baselines across six benchmarks, including IID and OOD settings. They also explore CoT reasoning but find it does not consistently improve editing quality.

**Questions:**

1. **Can RL discover better reasoning paths than pre‑collected CoTs?** If the off‑the‑shelf CoTs are suboptimal, can the RL stage improve not only editing quality but also generate superior chain‑of‑thought trajectories?
2. **Why omit a pure complex‑data SFT baseline?** The paper mixes simple and complex edits for SFT but does not show results when fine‑tuning only on the complex subset. How would SFT trained solely on complex examples perform?
3. **Best‑of‑N evaluation for SFT**: Figure 4 shows variability across samples, but only average scores are reported. Can the authors report best‑of‑N (e.g., best out of 5 or 10 samples) metrics for the SFT model?
4. **Best‑of‑N comparison between SFT and RL**: If best‑of‑N sampling is feasible, can the authors provide best‑of‑N results side‑by‑side for both SFT and the RL‑tuned model to isolate the benefit of policy optimization versus sampling strategy?

**Ethical Concerns:**

["NO or VERY MINOR ethics concerns only"]

**Final Justification:**

During the rebuttal phase, the author addressed my concern about best-of-N evaluation and explained the details about model choice. Given I assigned a high score, I'll maintain my score.

**Limitations:**

See weakness and questions.

**Quality:**

3

**Strengths And Weaknesses:**

## Strengths
- **Novelty**: Introduces a simple yet effective RL post‑training pipeline for autoregressive image editing, a relatively unexplored area.
- **Comprehensive evaluation**: Benchmarks on both simple (OmniEdit, EmuEdit) and complex (Aurora, MagicBrush, VisMin, I2EBench) tasks, with clear gains quantified via VIEScore.
- **Strong baseline comparison**: Matches or outperforms diffusion‑based methods (e.g., Omnigen) while using less data and provides a direct comparison to the only other autoregressive editing model, EditAR.
- **Insightful analysis**: The ablation on SFT data mixes and the investigation of CoT reasoning provide useful lessons about when and why each paradigm succeeds or fails.
- **Clarity and reproducibility**: Detailed descriptions of datasets, hyperparameters, and training settings support reproducibility.

## Weaknesses
- **Verifier dependency**: The approach hinges on a heavyweight MLLM reward model, which may be impractical for many users and introduces potential bias if the verifier’s understanding differs from human judgment.
- **Base model constraints**: Emu3’s limited image generation fidelity caps overall editing quality; stronger bases might change the relative benefits of RL vs. SFT.
- **Limited error analysis**: While average VIEScore gains are reported, detailed failure cases (e.g., semantic misalignments) are only qualitatively discussed, leaving some behaviors insufficiently characterized.
- **Scalability to high resolution**: All experiments use 256×256 images; it remains unclear how the pipeline scales to higher resolutions common in real‑world editing.

---

> ### Author Rebuttal · Authors · 2025-07-31
>
> Thank you for the detailed and positive review. We're glad you found our RL post-training pipeline **novel and effective**, our evaluation **comprehensive**, our experimental comparisons **strong**, our analysis of SFT data mixes and CoT reasoning **insightful**, and our paper **clear and reproducible**.
>
>
> >Verifier dependency: ... heavyweight MLLM reward model, ... impractical for many users
>
> We use Qwen2.5‑VL‑72B as the backend for VIEScore to provide high-quality reward signals during RL training. The full pipeline runs on a single 8×A100 node, which is feasible for many academic labs.
>
> >  ... potential bias if the verifier’s understanding differs from human judgment.
>
> **`VIEScore Evaluation with Qwen2.5‑VL‑72B and Alignment with Human Judgment `** To address the reviewer’s concern, we report the correlation between Qwen's judgement (as captured by VIEScore) and the human judgments on AURORA‑Bench (rows report the breakdown across subtasks and All denotes the aggregate performance). We see that these correlation values are in the same ballpark as those reported in the VIEScore paper (0.3821) [1] which established VIEScore as a reliable automatic evaluation metric. Thus these results validate the reliability of Qwen as a verifier. Also, the results show that Qwen is reliable not only on object and attribute centric edits but also on more complex edits such as spatial (WhatsUp) and action (Something‑Something, Action‑Genome).
>
>
> ### Correlation Results
>
> | Dataset    | Overall  | Semantic |
> |------------|----------|----------|
> | MagicBrush | 0.5377   | 0.6104   |
> | Something  | 0.3084   | 0.3192   |
> | AG         | 0.4385   | 0.4392   |
> | EPIC       | 0.2393   | 0.2886   |
> | Whatsup    | 0.4420   | 0.4556   |
> | All        | 0.3971   | 0.4309   |
>
> [1] VIEScore: Towards Explainable Metrics for Conditional Image Synthesis Evaluation. ACL 2024.
>
>
> >Base model constraints: Emu3’s limited image generation fidelity caps overall editing quality; stronger bases might change the relative benefits of RL vs. SFT.
>
>
> We agree that Emu3’s image fidelity places an upper bound on final editing quality. We chose Emu3 because it is currently the only open‑source autoregressive model that offers both sufficient image generation quality and semantic understanding of the prompts, and can be further fine‑tuned for the task of  image editing as well as optimized with RL (e.g., GRPO). Stronger AR base models are not available, making empirical comparisons infeasible at this time.
>
> Regarding the relative benefits of RL vs. SFT, while we are unable to run RL on stronger AR models directly, this trend in the literature supports our expectation that RL's advantages may be amplified as base model quality improves.
>
> Moreover, even with stronger base models, obtaining high-quality paired data for supervised fine-tuning—especially for complex edits like spatial rearrangement or action changes—remains a key bottleneck. In contrast, our RL approach does not rely on paired data and can generalize better to such tasks by leveraging verifier-based reward signals.
>
> Together, these factors suggest that RL remains essential and may offer increasing benefits as base model quality improves.
>
> >Limited error analysis: While average VIEScore gains are reported, detailed failure cases (e.g., semantic misalignments) are only qualitatively discussed, leaving some behaviors insufficiently characterized.
>
> To provide a clearer picture of model behavior, we break down VIEScore into its four aspects: edit success and over-edit (capturing semantic alignment), and naturalness and absence of artifacts (capturing perceptual quality). We report the scores for each of these aspects comparing SFT(S) and our best-performing setup, SFT(S) → RL(S+C). The results in the AVERAGE column show that RL significantly improves the performance on Edit Success, indicating stronger semantic alignment with the edit instructions, without causing significant drop in the other aspects such as OverEdit, and perceptual quality metrics such as Looks Natural and No Artifact.
>
>
> **SFT(S) vs. SFT(S) → RL(S+C)**
> (Values are shown as `SFT(S) / SFT(S) → RL(S+C)`, with the higher score in bold)
>
> | Metric          | AURORA       | MagicBrush    | OmniEdit     | I2EBench     | Vismin       | EmuEdit      | AVERAGE     |
> |-----------------|--------------|---------------|--------------|--------------|--------------|--------------|-------------|
> | **Edit Success**   | 3.0 / **3.6** | 2.6 / **4.1** | 6.1 / **7.0** | 3.7 / **4.3** | 3.2 / **4.3** | 3.5 / **4.2** | 3.7 / **4.6** |
> | **OverEdit**       | **7.0** / 6.6 | **7.8** / 7.0 | 7.3 / **7.5** | 6.8 / **6.9** | **5.7** / 6.5 | **7.5** / 7.0 | **7.0** / 6.9 |
> | **Looks Natural**  | **5.8** / 5.7 | **6.3** / 5.8 | 6.7 / **6.9** | 5.0 / **5.1** | 6.6 / **6.8** | **6.3** / 6.0 | **6.1** / 6.0 |
> | **No Artifact**    | **6.8** / 6.7 | **6.9** / 6.4 | 7.5 / **7.7** | 5.8 / **5.9** | 7.3 / **7.6** | **6.9** / 6.7 | **6.9** / 6.8 |
> | **VIEScore**       | 3.6 / **4.0** | 3.2 / **4.3** | 5.7 / **6.3** | 3.6 / **4.1** | 3.6 / **4.5** | 3.7 / **4.3** | 3.9 / **4.6** |
>
>
>
> >Scalability to high resolution: All experiments use 256×256 images; it remains unclear how the pipeline scales to higher resolutions common in real‑world editing.
>
> Due to the limited resources as an academic lab, we conducted all experiments at 256×256 resolution. However, the method is fully scalable to higher resolutions given sufficient compute. Scaling does not require any architectural changes—only minor adjustments for tokenizing higher-resolution images during training. We also note that prior works such as InstructPix2Pix, MagicBrush, and Aurora, all developed in academic labs, were trained at the same 256x256 resolution.
>
>
>
> >Can RL discover better reasoning paths than pre‑collected CoTs? If the off‑the‑shelf CoTs are suboptimal, can the RL stage improve not only editing quality but also generate superior chain‑of‑thought trajectories?
>
> In our experiments, we did not observe clear emergent changes in the structure of the reasoning chains after RL post‑training. A possible explanation is that the pretrained model was not explicitly optimized for reasoning or chain‑of‑thought style supervision, which limited the scope for RL to substantially modify or restructure such trajectories. As a result, RL mainly improved editing quality without noticeably altering the reasoning style.
>
> >Why omit a pure complex‑data SFT baseline? The paper mixes simple and complex edits for SFT but does not show results when fine‑tuning only on the complex subset. How would SFT trained solely on complex examples perform?
>
> We trained an SFT model only on complex edits, but the available datasets are small for complex edits; for example, MagicBrush and ActionGenome each have only about 8K examples. That’s not enough to build a strong, balanced training set, especially since the Emu3 base model first needs to learn the task of editing (its pretraining does not cover image editing task, only image generation task). To adapt it for editing and achieve decent editing performance, we need a sufficiently large dataset.
>
> However, to address the reviewer’s concern, we trained a model (EARL SFT(C)) using the available complex subset and present the results here, compared against EARL  SFT(S):
>
> **Comparison of Baselines: SFT(S) vs. SFT(C)**
>
> | Metric | AURORA | MagicBrush | OmniEdit | I2EBench | Vismin | EmuEdit | AVERAGE |
> |--------|---------|------------|----------|----------|--------|---------|---------|
> | EARL – SFT(S) | 3.58 | 3.19 | 5.73 | 3.59 | 3.57 | 3.66 | 3.88 |
> | EARL – SFT(C) | 2.78 | 2.71 | 2.93 | 2.67 | 3.90 | 2.68 | 2.94 |
>
> EARL – SFT(C) performs much worse than EARL – SFT(S), even on complex benchmarks. This confirms that with the limited complex-only data, the model fails to effectively learn the complex editing tasks.
>
>
> >Best‑of‑N evaluation for SFT: Figure 4 shows variability across samples, but only average scores are reported. Can the authors report best‑of‑N (e.g., best out of 5 or 10 samples) metrics for the SFT model?
>
> >Best‑of‑N comparison between SFT and RL: If best‑of‑N sampling is feasible, can the authors provide best‑of‑N results side‑by‑side for both SFT and the RL‑tuned model to isolate the benefit of policy optimization versus sampling strategy?
>
> Thank you for the insightful suggestion. We ran best‑of‑N evaluations for SFT(S) and SFT(S) → RL(S+C) to separate sampling gains from policy optimization. Results are shown below.
>
>
> ### Best‑of‑N Evaluation Results
>
> | Model                         | AURORA | MagicBrush | OmniEdit | I2EBench | Vismin | EmuEdit | AVERAGE |
> |-------------------------------|--------|------------|----------|----------|--------|---------|---------|
> | EARL SFT (S)                  | 3.58   | 3.19       | 5.73     | 3.59     | 3.57   | 3.66    | 3.88    |
> | EARL SFT (S) [Best_of_5]      | 3.89   | 3.63       | 6.13     | 4.00     | 4.09   | 4.21    | 4.33    |
> | EARL SFT (S) → RL (S+C)       | 3.99   | 4.26       | 6.33     | 4.08     | 4.48   | 4.28    | 4.57    |
> | EARL SFT (S) → RL (S+C) [Best_of_5] | **4.08** | **4.66** | **6.54** | **4.36** | **4.69** | **4.66** | **4.83** |
>
>
>
> We observe that:
> **SFT(S) with vs. without best-of-5:**  Best‑of‑5 results are significantly higher than single‑sample results, indicating substantial room for improvement through policy optimization, thus further motivating our RL post-training approach.
>
>
> **SFT(S)-with best-of-5 vs. SFT (S) → RL (S+C):** RL outperforms SFT-with best‑of‑5-sampling, showing that **RL training provides improvements beyond what can be achieved via sampling alone**.
>
>
> **SFT (S) → RL (S+C) with vs. without best-of-5:** Best‑of‑5 on top of the RL post-trained model gives the highest performance overall. The gap between RL single‑sample and RL best‑of‑5 shows how close our RL model is to an “ideal” RL‑trained policy. We see that the gap is quite small, establishing the effectiveness of our trained policy.

---

> ### Comment · Reviewer_vNkF · 2025-08-04
>
> Thank you for your response. Using best-of-N is indeed a reasonable way to highlight the limitations of SFT and the potential of RL. However, I feel that my concern has not been fully addressed, for example, regarding the limited base model evaluation. Have you considered using a stronger base model such as Janus-Pro [1] for experiments?
>
> [1]: https://huggingface.co/deepseek-ai/Janus-Pro-1B

---

> > ### Author Response · Authors · 2025-08-05
> > **Reply to Reviewer vNkF**
> >
> > First, we thank the reviewer for suggesting the best‑of‑N experiment. We believe this significantly strengthened the paper by clarifying the motivation for RL.
> >
> > >Have you considered using a stronger base model such as Janus-Pro 1 for experiments?
> >
> > Yes, we initially explored the Janus Pro models. To the best of our knowledge, there is currently no stronger autoregressive (AR) model than Emu3 that is suitable for the task of **image editing**. We considered Janus Pro but found it unsuitable due to its architecture. Unlike Emu3 and other competitive image editing models such as EditAR [1], InstructPix2Pix [2], and OmniGen [3], Janus’s image encoder is decoupled from its decoder, which makes it less effective for image editing. In contrast, Emu3 and similar models encode input images with the same VAE tokenizer used by the decoder for generation, thereby preserving both semantics and structure and enabling faithful, localized edits. Janus, however, relies on a SigLIP encoder that captures only high‑level semantics while discarding pixel‑level details and structural information [4]. As a result, its conditional image representations cannot reliably preserve structure for localized changes or maintain the fine‑grained detail of the input image needed for image editing. This loss of pixel‑level and layout information makes Janus unsuitable for preserving the original input during editing. Moreover, in our initial qualitative evaluation of image generation, we also found that Emu3 produces higher‑quality natural images on COCO‑like datasets [5] compared to Janus.
> >
> >
> >
> >
> > [1] EditAR: Unified Conditional Generation with Autoregressive Models. arXiv preprint arXiv:2501.04699, 2025.
> >
> > [2] InstructPix2Pix: Learning to Follow Image Editing Instructions. arXiv preprint arXiv:2211.09800, 2022.
> >
> > [3] OmniGen: Unified Image Generation. arXiv preprint arXiv:2409.11340, 2024.
> >
> > [4] CLIP Behaves like a Bag‑of‑Words Model Cross‑modally but not Uni‑modally, arXiv preprint arXiv:2502.03566.
> >
> > [5] Microsoft COCO: Common Objects in Context. arXiv preprint arXiv:1405.0312, 2015.

---

> > > ### Comment · Reviewer_vNkF · 2025-08-05
> > >
> > > Thank you for your additional explanation. I'll maintain my score, given that I already assigned a high score to this work.

---

### Official Review · Reviewer_EDq6 · 2025-06-29

**Clarity:** 3
**Significance:** 3
**Originality:** 2
**Rating:** 4
**Confidence:** 3

**Summary:**

The paper introduces EARL (Editing with Autoregression and RL), a two-stage framework for text-guided image editing. Stage 1 performs standard teacher-forced fine-tuning on 750 K “simple” and 300 K “complex” edit pairs; Stage 2 optimises the same network with GRPO, drawing rewards from Qwen-VL via VIEScore’s four sub-metrics (edit success, over-edit, naturalness, artefacts). Evaluations on OmniEdit, VisMin, I2E-Bench and three in-house splits show that RL lifts the average VIEScore, edging out diffusion-based MagicBrush and Aurora but still trailing OmniEdit.

**Questions:**

None

**Ethical Concerns:**

["NO or VERY MINOR ethics concerns only"]

**Final Justification:**

This paper is well-written and easy to follow, and with most of my initial questions addressed in the rebuttal, I will maintain my rating.

**Limitations:**

yes

**Paper Formatting Concerns:**

None.

**Quality:**

3

**Strengths And Weaknesses:**

Pros:

1.The prose is clean and easy to follow, with all key stages (SFT, RL, evaluation) described in a logical order.

2.Unifies simple and complex edits in a single autoregressive transformer.

3.Beats diffusion baselines MagicBrush/Aurora on average VIEScore without diffusion back-bones.

Cons:

1.Over-stated novelty:

(1)“First RL for autoregressive editing” ignores prior EditAR and InsightEdit work.

(2)GRPO application to vision is incremental; similar RLHF tricks already common in LLM papers.

2.Evaluation gaps & reward leakage

(1)Same Qwen-VL judge is used both as the RL reward and as the test metric, inviting reward-hacking.

(2)No cross-metric checks (e.g., CLIP-SIM, FID) and zero human preference studies.

3.Missing baselines
(1)Lacks head-to-head numbers against EditAR, EmuEdit (XL), OmniGen diffusion + RL, and other recent autoregressive editors.

---

> ### Author Rebuttal · Authors · 2025-07-31
>
> Thank you for the constructive review. We are glad you found the **prose clean and easy to follow**, and we appreciate your **recognition of the paper’s logical structure across SFT, RL, and evaluation**. We're also encouraged by your **appreciation of our effort to unify simple and complex edits in a single autoregressive transformer**, and the **model’s ability to beat diffusion baselines like MagicBrush and Aurora on average VIEScore without relying on diffusion backbones**.
>
> >Evaluations on OmniEdit, VisMin, I2E-Bench and three in-house splits show that RL lifts the average VIEScore, edging out diffusion-based MagicBrush and Aurora but still trailing OmniEdit.
>
>
> We believe the reviewer is referring to Omnigen (the model), since OmniEdit is one of the benchmarks we evaluate on, where we already outperform the other baselines.
>
> **`EARL Scaled RL Post‑Training`**  Post submission, we further scaled up the duration of the RL training for our best-performing setup (SFT(S) → RL(S+C)) to 2000 steps with 32k samples (vs. 1000 steps and 1.6k samples).
>
>
> | Model   | OmniEdit | EmuEdit | AURORA | MagicBrush | VisMin | I2EBench | AVG  |
> |---------|----------|---------|--------|------------|--------|----------|------|
> | EARL    | **6.39** | 4.47    | **4.27** | 4.52       | **4.93** | 4.19     | **4.80** |
> | Omnigen | 5.68     | **5.00** | 4.10   | **4.68**   | 4.09   | **4.68** | 4.70 |
>
> We see that with scaling-up, EARL now outperforms even Omnigen on AVG, despite using ~5× less image editing data (782k image editing sample for the scaled up EARL vs 3.7M image editing samples for Omnigen), demonstrating the effectiveness and scalability of our approach.
>
> >1.Over-stated novelty: (1)“First RL for autoregressive editing” ignores prior EditAR and InsightEdit work. (2)GRPO application to vision is incremental; similar RLHF tricks already common in LLM papers.
>
> 1- While EditAR is an autoregressive model, it does not use reinforcement learning, and InsightEdit relies on diffusion-based generation rather than autoregressive methods—and also does not include RL in its pipeline. To our knowledge, EARL is the first to apply reinforcement learning to autoregressive image editing, a combination not explored in prior work.
>
> 2- While RL has been explored in NLP and diffusion-based image generation, applying RL to autoregressive (AR) image editing is both novel and underexplored, as also acknowledged by R3: “Novelty: Introduces a simple yet effective RL post-training pipeline for autoregressive image editing, a relatively unexplored area.” The rapid improvements in image editing tasks have been led by diffusion‑based approaches, which have outpaced autoregressive models [1, 2]. Our work challenges this by showing that applying RL post-training to AR models significantly boosts performance, making them competitive with diffusion-based models while being more data-efficient.
>
> Moreover, our work is the first to conduct a systematic analysis of SFT vs RL for image editing. Through comprehensive experiments, we show that supervised fine-tuning (SFT) is not sufficient for achieving strong performance on complex edits such as spatial rearrangement, counting, and actions due to the lack of high quality and large scale paired datasets for such complex edits (see Table 1 for the sizes of available complex datasets). In contrast, RL post-training with a strong VLM verifier **does not require paired data** and **significantly helps improve the performance on complex edits**. Our strongest model SFT(S) → RL(S+C) significantly outperforms SFT on complex edits.
>
>
> [1] Magicbrush: A manually annotated dataset for instruction-guided image editing. Advances in Neural Information Processing Systems
>
> [2] Instructpix2pix: Learning to follow image editing instructions. In Proceedings of the IEEE/CVF Conference on Computer Vision and Pattern Recognition
>
>
> > 2.Evaluation gaps & reward leakage (1)Same Qwen-VL judge is used both as the RL reward and as the test metric, inviting reward-hacking. (2)No cross-metric checks (e.g., CLIP-SIM, FID) and zero human preference studies.
>
>
> 1- This is a misunderstanding—**we do not use the same model as the reward model and as the evaluation metric**. As described in Section 4.2, we use Qwen2.5-VL-72B as the RL reward model and GPT-4o-mini as the evaluation metric. This separation was intentional to avoid reward hacking and to ensure fair evaluation.
>
> 2- We chose **VIEScore as it shows stronger Metric-to-Human (M-H) correlations compared to traditional metrics** [3]. For example, in text-guided image editing, VIEScore (0.3821) outperforms LPIPS [4] (0.1142), and in subject-driven image editing, VIEScore (0.4637) surpasses CLIP-I [5] (0.3058), demonstrating closer alignment with human judgment. Our **human studies further confirm this**, showing high alignment between VIEScore (0.4328) and human preferences (**see Appendix C**: VIEScore Evaluation with GPT-4o-Mini and Alignment with Human Judgment).
>
>
> Also, in Section 5.2, we compare EARL with EditAR using a broader set of metrics beyond VIEScore—including PSNR, LPIPS, SSIM, and CLIP similarity—to provide a more complete evaluation.
>
> As for FID, we note that VIEScore already captures key aspects such as image realism and artifacts—core elements FID is designed to assess. However, to address the reviewer’s concern, we report FID and CLIP similarity scores between the ground truth and generated images averaged across all benchmarks that provide the ground truth edited image (MagicBrush, VisMin and OmniEdit).
>
>
>
>
>
>
> ### CLIP Similarity (ViT-B/32) – Edited Image vs. GT Edited Image (↑ higher is better)
> | Baseline              | MB   | VisMin | OmniEdit | AVG  |
> |---------------------|------|--------|----------|------|
> | Magicbrush          | 0.93 | 0.91   | 0.88     | 0.91 |
> | InstructPix2Pix     | 0.86 | 0.86   | 0.85     | 0.85 |
> | Aurora              | 0.92 | 0.90   | 0.86     | 0.89 |
> | Omnigen             | 0.92 | 0.92   | 0.88     | 0.90 |
> | EditAR              | 0.87 | 0.91   | 0.84     | 0.88 |
> | EARL SFT(S)→RL(S+C) | 0.88 | 0.88   | 0.89     | 0.88 |
>
>
> ### FID (InceptionV3) – Edited Images vs. GT Edited Images (↓ lower is better)
> | Method              | MB    | VisMin | OmniEdit | AVG   |
> |---------------------|-------|--------|----------|-------|
> | Magicbrush          | 25.70 | 42.55  | 53.52    | 40.59 |
> | InstructPix2Pix     | 46.03 | 53.26  | 60.30    | 53.20 |
> | Aurora              | 32.89 | 48.05  | 60.71    | 47.22 |
> | Omnigen             | 37.84 | 40.30  | 57.53    | 45.22 |
> | EditAR              | 46.68 | 39.76  | 61.97    | 49.47 |
> | EARL SFT(S)→RL(S+C) | 47.41 | 47.16  | 54.34    | 49.64 |
>
> We see that EARL’s scores remain in the same range as Omnigen, Aurora and EditAR for both FID and CLIP, showing it preserves semantic alignment and image quality at a comparable level to other strong baselines.
>
>
> [3]  VIEScore: Towards Explainable Metrics for Conditional Image Synthesis Evaluation. arXiv preprint arXiv:2312.14867.
>
> [4] The unreasonable effectiveness of deep features as a perceptual metric. CVPR, 2018.
>
> [5]  Learning Transferable Visual Models From Natural Language Supervision. arXiv preprint arXiv:2103.00020.
>
>
> >  3.Missing baselines (1)Lacks head-to-head numbers against EditAR, EmuEdit (XL), OmniGen diffusion + RL, and other recent autoregressive editors.
>
>
>
> We already include comparisons with EditAR in our evaluation (Table 3). Since EditAR was not open-sourced at the time of submission, we could not  additionally report its performance on all the other benchmarks we have in our paper (Table 2) However, now it’s opensource, so we have run these additional evaluations. In our best-performing setup (SFT(S) → RL(S+C)), EARL achieves 4.57 versus EditAR’s 4.20 average VIEScore across all benchmarks, demonstrating superior performance. We will update the paper with the new EditAR results.
>
>
>
> | Baseline              | AURORA | MagicBrush | OmniEdit | I2EBench | VisMin | EmuEdit | AVERAGE |
> |------------------------|--------|------------|----------|----------|--------|---------|---------|
> | EditAR                | 3.79   | 3.84       | 5.29     | 3.84     | 4.54   | 3.88    | 4.20    |
> | EARL SFT(S)→RL(S+C)   | 3.99   | 4.26       | 6.33     | 4.08     | 4.48   | 4.28    | 4.57    |
>
> As for EmuEdit (XL), to the best of our knowledge it is not publicly available. Lastly, as for OmniGen diffusion + RL, to the best of our knowledge, no prior work combines OmniGen (a diffusion based model) with reinforcement learning. Our goal is to demonstrate that autoregressive models are a strong alternative to diffusion-based models. Exploring reinforcement learning for OmniGen is an interesting direction, but it is outside the scope of this work.

---

> > ### Author Response · Authors · 2025-08-06
> > **Reply to Reviewer EDq6**
> >
> > Thank you again for your thoughtful feedback. We hope our rebuttal has adequately addressed your concerns. If there are any remaining questions or points that could benefit from further clarification, we would be happy to provide additional details. We appreciate your time and consideration.

---

> > > ### Comment · Reviewer_EDq6 · 2025-08-09
> > >
> > > Thank you for your additional explanation. I'll maintain my score, given that I already assigned a high score to this work.

---

### Official Review · Reviewer_W5tw · 2025-07-01

**Clarity:** 4
**Significance:** 2
**Originality:** 2
**Rating:** 4
**Confidence:** 4

**Summary:**

The paper introduces the Editing with Autoregression and RL (EARL) model, which integrates autoregressive generation with reinforcement learning (RL) to enhance image editing tasks. Leveraging a multimodal large language model (MLLM) verifier, such as Qwen2.5-VL-72B, EARL optimizes the editing process by refining the model's outputs based on feedback from the verifier. The authors compare this approach with other strategies, including supervised fine-tuning (SFT) and chain-of-thought (CoT) reasoning. Experimental results demonstrate that EARL outperforms diffusion model baselines in both simple and complex editing tasks, offering higher data efficiency.

**Questions:**

1. When using Qwen2.5-VL-72B as a reward model, has its scoring preference for different types of edits been validated?

2. The paper mentions that EARL "uses less data," but it does not quantify the differences in training time and GPU memory usage compared to other methods. For example, under the same hardware conditions, how many times longer does the total time of EARL's SFT+RL process take compared to the diffusion model?

3. The paper discusses the potential of combining autoregressive models with RL in image editing but does not examine the potential drawbacks of their token-by-token generation mechanism for complex edits (such as global semantic consistency). For example, when processing the instruction "swap the positions of two objects in the image," does the autoregressive model introduce spatial reasoning errors due to the pixel-level generation order?

**Ethical Concerns:**

["NO or VERY MINOR ethics concerns only"]

**Final Justification:**

This paper is well-written and easy to read. The proposed method, which integrates autoregressive models and reinforcement learning (RL) for image editing, represents a novel contribution to the field. While I raised some questions during the initial review stage, the authors have effectively addressed most of these concerns in their rebuttal. I thus believe this paper meets the acceptance criteria.

**Limitations:**

Qwen2.5-VL-72B, as a validator, may have training data biases (such as preferences for certain object categories or language expressions), but the paper does not analyze the impact of these biases on the editing results (such as whether they lead to over-editing or under-editing in specific scenarios).

**Quality:**

2

**Strengths And Weaknesses:**

Strengths：

1. The authors compare supervised fine-tuning (SFT), reinforcement learning (RL), and chain-of-thought (CoT) reasoning within a unified autoregressive framework. They find that RL significantly enhances complex editing tasks, such as spatial transformations and action understanding, whereas CoT does not lead to consistent improvements. This underscores the clear motivation for integrating autoregressive generation with RL.

2. The experimental validation encompasses a mixed dataset of simple and complex editing tasks, evaluating model generalization across six benchmarks, including OmniEdit and Aurora. Notably, the model demonstrates strong competitiveness in out-of-distribution (OOD) scenarios.

3. This paper is well-written and easy to read.

Weaknesses：

1. This paper proposes integrating autoregressive models with RL for image editing. However, similar approaches have been explored in the NLP domain, such as RLHF applied to large language models (LLMs). Additionally, the application of RL in image generation, exemplified by methods such as Diffusion-DPO and InstructRL4Pix, is not particularly novel. The core innovation of EARL lies more in the integration of RL within an autoregressive framework, representing a model architecture combination rather than a technical breakthrough.

2. The autoregressive token-by-token generation mechanism of autoregressive models may result in slower inference speeds compared to diffusion models. However, the paper does not provide a comparison of inference times between EARL and other AR or diffusion-based models such as EditAR (CVPR  25) and MagicBrush (NeurIPS 23), making it difficult to assess its potential for real-time applications.

3. This paper employs Qwen2.5-VL-72B as the sole RL verifier, without comparing its performance against other multimodal large language models (MLLMs) such as LLaVA and MiniGPT-4. This limitation hinders the assessment of the verifier's impact on the overall results.

4. The paper does not provide a quantitative comparison of training times and GPU memory usage between EARL and other models. This omission hinders a comprehensive evaluation of EARL's data efficiency advantages.

---

> ### Author Rebuttal · Authors · 2025-07-31
>
> Thank you for the thoughtful review. We're glad you found the paper **well-written** and **easy to read**. We appreciate the **recognition of our unified comparison** of SFT, RL, and CoT, and the **clear motivation for integrating autoregressive generation with RL**. We're also encouraged by the **acknowledgment of our comprehensive evaluation** and the model’s **strong competitiveness in OOD scenarios**.
>
> > ...  similar approaches have been explored in the NLP domain, …  Diffusion-DPO and ,... The core innovation of EARL …, representing a model architecture combination rather than a technical breakthrough
>
> While RL has been explored in NLP and diffusion-based image generation, applying RL to autoregressive (AR) image editing is both novel and underexplored, as also acknowledged by R3: “Novelty: Introduces a simple yet effective RL post-training pipeline for autoregressive image editing, a relatively unexplored area.” The rapid improvements in image editing tasks have been led by diffusion‑based approaches, which have outpaced autoregressive models [1, 2] . Our work challenges this by showing that applying RL post-training to AR models significantly boosts performance, making them competitive with diffusion-based models while being more data-efficient.
>
> Moreover, our work is the first to conduct a systematic analysis of SFT vs RL for image editing. Through comprehensive experiments, we show that supervised fine-tuning (SFT) is not sufficient for achieving strong performance on complex edits such as spatial rearrangement, counting, and actions due to the lack of high quality and large scale paired datasets for such complex edits (see Table 1 for the sizes of available complex datasets). In contrast, RL post-training with a strong VLM verifier **does not require paired data** and **significantly helps improve the performance on complex edits**. Our strongest model SFT(S) → RL(S+C) significantly outperforms SFT on complex edits.
>
> Post‑submission, we ran **scaled RL post‑training** in our best setup (SFT(S) → RL(S+C)), which **outperformed Omnigen** with an average score of 4.8 vs. 4.7 across benchmarks. Please see our response **`EARL Scaled RL Post‑Training`** to **Reviewer 2**  for details.
>
>
> [1] MagicBrush. arXiv preprint arXiv:2306.10012.
>
> [2] Instructpix2pix. arXiv preprint arXiv:2211.09800.
>
> > The autoregressive token-by-token generation … may result in slower inference speeds compared to diffusion models. …  potential for real-time applications.
>
> We have integrated EARL with vLLM [3], which enables fast autoregressive decoding via PagedAttention [3] and optimized parallel token generation. This significantly reduces latency and improves throughput over standard AR decoding.
>
> We evaluated inference speed on 50 samples using a single A100L GPU. All models were run at 256×256 resolution, except EditAR which was trained and tested at 512×512. Reported times are the total time to generate 50 samples. EARL: 52.7 s, EditAR: 66.19 s, MagicBrush: 23.6 s, InstructPix2Pix: 23.7 s, Aurora: 23.7 s, Omnigen: 200 s. We make the following observations regarding inference speed:
>
> **EARL vs. Omnigen.** EARL is roughly **4× faster than Omnigen** while achieving competitive editing performance (see Table 2), highlighting the benefit of autoregressive generation with optimized decoding.
>
>
> **EARL vs. weaker diffusion baselines.** EARL is about **2× slower than MagicBrush, InstructPix2Pix, and Aurora**, but we believe this **trade-off is justified by the significant improvements in editing quality** (Table 2).
>
>
> **Clarifying EditAR results.** Since EditAR was trained on 512×512 images, we reported its runtime at that resolution (66.19s for 50 samples). EditAR generates higher-resolution outputs, so its runtime naturally is higher.  At 256×256, it produces noisy samples (46.30s for 50 samples).
>
> [3] Efficient Memory Management for Large Language Model Serving with. Proceedings of the ACM SIGOPS 29th Symposium on Operating Systems Principles.
>
>
> >This paper employs Qwen2.5-VL-72B as the sole RL verifier, without comparing … (MLLMs) such as LLaVA and MiniGPT-4. This limitation hinders the assessment of the verifier's impact on the overall results.
>
> We appreciate the reviewer’s suggestion regarding alternate verifiers. We selected **Qwen2.5-VL-72B** because it demonstrates strong performance on fine-grained vision-language tasks [1], comparable to GPT-4o, making it a reliable source of reward signals. In contrast, earlier VLMs such as **LLaVA** and **MiniGPT-4** **struggle** with **instruction-following and prompt adherence**, and in our experiments their outputs were too noisy to extract usable rewards.
>
> To further validate the reviewer’s point, we also tested a more recent **Qwen-7B** model, which is instruction-tuned and closer in scale to LLaVA and MiniGPT-4 (~7B). However, we observed that the performance of our model degrades significantly when using such a weaker verifier. For example, under our best-performing setup (SFT(S) → RL(S+C)), the AR model trained on Qwen‑7B achieved an average score of **2.81**, compared to **3.88** with SFT alone, showing clear degradation. This confirms that smaller models cannot provide sufficiently accurate or stable feedback for AR-based RL training, underscoring the need for a stronger verifier such as Qwen2.5-VL-72B. Given the limited rebuttal period, we were not able to perform RL post-training with other families of MLLMs. We plan to include these results in the camera-ready version.
>
>
>
> | Baseline                                     | AURORA | MagicBrush | OmniEdit | I2EBench | Vismin  | EmuEdit | AVERAGE |
> |----------------------------------------------|--------|------------|----------|----------|---------|---------|---------|
> | **SFT (S)**                                  | 3.58   | 3.19       | 5.73     | 3.59     | 3.57    | 3.66    | 3.88    |
> | **SFT (S) → RL (S+C)— Qwen7B** | 2.55  | 2.66      | 4.05    | 2.60    | 2.32  | 2.69  | 2.81   |
>
>
> [1] Qwen2.5 Technical Report. arXiv:2412.15115.
>
> > … quantitative comparison of training times and GPU memory usage between EARL and other models. …
> 	>... EARL "uses less data," .. quantify the differences in training time.  …
>
>  We compare the training time, GPU usage, and data size for EARL and available baselines below:
>
>
> | Model           | Training Data                                                                        | Compute Resources                        | Duration (Hours)                       |
> |-----------------|--------------------------------------------------------------------------------------|------------------------------------------|-----------------------------------------|
> | **EARL (ours)** | ~752k samples   (SFT: ~750k, RL: 1.6k)                                     | 8×A100L   | 84h (SFT: ~60h,RL: ~24h) |
> | InstructPix2Pix | 313k samples                                                                         | 8×A100                              | 25.5h                                   |
> | MagicBrush      | ~10k samples   (built on InstructPix2Pix)                                          | 2×A100              | – (Not reported in the paper)         |
> | Aurora          | 289k samples   (built on InstructPix2Pix)                                        | 2× RTX A6000                              | 16h                                    |
> | Omnigen         |~0.1B samples, including ~3.7M image editing examples                                                         | 104×A800                                 |  – (Not reported in the paper)                     |
> | EditAR          | 1.5M samples                                                | 8×A100                                   | – (Not reported in the paper)     |
>
> As seen, EARL achieves competitive performance using ~5× less image editing data (752k vs 3.7M image editing samples) and far fewer GPUs (8 vs. 104) than Omnigen, highlighting its superior data and training efficiency. The higher GPU usage in OmniGen is expected, since their model is trained from scratch. In contrast, EARL builds on pretrained models, which naturally reduces computational requirements. This comparison suggests that EARL provides practical efficiency benefits while maintaining competitive performance.
>
> > When using Qwen2.5-VL-72B as a reward model, has its scoring preference for different types of edits been validated?
>
> > Qwen2.5-VL-72B, as a validator, may have training data biases …
>
> Qwen2.5‑VL‑72B, as a reward model, may indeed reflect biases from its training data. While we did not conduct a full bias analysis, we provide evidence in Appendix Section D showing some inconsistencies, particularly on fine‑grained edits such as counting or spatial relations. Importantly, we validated its reliability by measuring correlation with human judgments on AURORA‑Bench (see **`VIEScore Evaluation with Qwen2.5‑VL‑72B and Alignment with Human Judgment`**, response to **Reviewer 3**). These results show strong agreement with human ratings across diverse edit types, supporting its use as a reward signal.
>
>
> > The paper discusses the potential of combining autoregressive models with RL in image editing but … potential drawbacks of their token-by-token generation …
>
> Token-by-token generation in autoregressive models can lead to error propagation and accumulation, especially in SFT-only settings, where early generation mistakes compound over time and introduce artifacts. In contrast, our RL-trained models show significantly fewer artifacts, as the verifier provides strong reward signals that encourage globally coherent and artifact-free outputs. As shown in Appendix Section E, performance breakdown across different edit types (Tables 2-6) shows consistent improvements in edit quality across diverse edit types such as counting, spatial relations, etc. This indicates that the verifier’s reward signal helps guide the model toward more accurate and coherent edits.

---

> > ### Comment · Reviewer_W5tw · 2025-08-03
> > **reply to rebuttal**
> >
> > I thank the authors for their answers to my questions and discussion points. Having read the other reviews and rebuttals as well, most of my initial concerns are addressed.

---

> > > ### Author Response · Authors · 2025-08-03
> > > **Reply to Reviewer W5tw**
> > >
> > > Thank you for initiating this valuable discussion. It has been helpful in refining our work. We are happy to address any remaining concerns and hope our response has contributed to a more positive reassessment of the work.

---

### Note · Authors · 2025-08-13

We thank all reviewers for their thoughtful reviews, constructive questions, and engagement. Your feedback, proposed experiments, and requests for clarification have greatly improved our paper, and we will integrate all comments into the camera-ready version. We are pleased by the overall positive assessment and glad that our responses addressed reviewers' concerns, with no additional issues remaining.

## Main points addressed during the rebuttal period:
- **Novelty/contributions (RW5tw, REDq6)**: Clarified that applying RL to autoregressive (AR) image editing is novel, as progress so far has been driven by diffusion-based methods, leaving AR models underexplored. Our method makes AR editing competitive with strong diffusion baselines and offers the first systematic comparison of SFT, RL, and CoT for image editing.
- **Comparison with EditAR (REDq6, RWQB5)**: EditAR was not open-source at submission time, so only PIEBench results were available. Now, we report results on the full benchmark, outperforming EditAR on average VIEScore.
- **Reward model**: Addressed REDq6’s misunderstanding by clarifying that the reward model and the evaluation judge are not the same. In response to RvNkF’s concern about MLLM-based reward models, we clarified that it is practical for academic labs and aligns well with human preferences.
- **Evaluation requested:**
  - RW5tw: Provided training and inference time comparisons, showing our model is ~4× faster than the best editing model baseline,  Omnigen. We also included results using a small-sized verifier model as requested.
  - RvNkF: Added the complex-data SFT baseline (SFT-C) and performed Best-of-N evaluation for both SFT and RL, highlighting RL’s performance gains. We also conducted a quantitative error analysis of failure cases to identify improvement areas.
  - REDq6: Included additional evaluation metrics, specifically CLIP and FID, to broaden the assessment.
  - In addition, we include post-submission scaled RL results, which outperform the strongest existing baseline.
- **Choice of base model and RL method (RvNkF, RWQB5):** Clarified that Emu3 is chosen as the base model because it is the most suitable AR model for image editing, though we initially explored Janus. We chose GRPO for its proven stability and efficiency in large-scale AR training, aiming to establish RL as a viable approach for AR image editing.

We appreciate the constructive dialogue and look forward to presenting the improved version of this work.

---

### Decision · Program_Chairs · 2025-09-17

**Decision:**

Accept (poster)

**Comment:**

This paper introduces the Editing with Autoregression and RL (EARL) model. Initially, the reviewers raised concerns about the lack of inference-time comparisons between EARL and other AR or diffusion-based models, limited key novelty, verifier dependency, and insufficient discussion of experimental results. However, after the rebuttal, the authors successfully addressed most of these concerns, including clarifications regarding novelty, and all reviewers provided positive feedback. The AC has carefully reviewed the paper, the reviewer comments, and the rebuttal, and agrees that the paper is well-motivated, clearly written, and supported by thorough experiments. Therefore, the AC recommends acceptance. It is encouraged that the authors incorporate all additional experiments and discussions from the rebuttal into the final version.